# "Mills of God": Two Ways of Envisaging Justice and Punishment in Greek Antiquity

**Duluo Nie**

School of International Studies, Sun Yat-sen University, Zhuhai 519082, China; nieduluo@gmail.com

**Abstract:** This paper discusses two typical Greek traditions of envisaging punishments for wrongdoings: one is the religious idea of inherited responsibility, and the other is the invention and evolution of the notion of hell. The former idea, sometimes summarized by authorities such as Gustave Glotz, Eric Dodds, and Hugh Lloyd-Jones under the terms inherited guilt, ancestral fault, and responsabilité héréditaire, is one of the major themes running through the writings of authors of both the Archaic and Classical periods, and is found in genres such as elegy, historiography, oratory, and prominently tragedy. As a core idea of Greek literature, it suggests that the descendants of wrongdoers are punished not for their own sins but for those of their ancestors. With the exclusion of ideas of a punishing hell, an afterlife, and the transmigration of souls, the doctrine of inherited responsibility has its own necessity for sustaining belief in the efficacy of divine punishment, given the common human experience that evil generally escapes punishment. Solon is the first Greek author to make such a statement explicitly. The latter tradition has a much longer history, which runs from Homer to Plato. Nonetheless, the descriptions of hell from Homer onwards do not remain consistent and uniform. Its evolution with the gradual incorporation of religious ideas such as afterlife punishment and transmigration of souls witnesses the need for a much more self-sufficient interpretation of cosmic justice than the notion of inherited responsibility. One interesting fact about the two traditions is that both have coexisted in the same period of time in the testimony of contemporary authors and even in the same author, notably Herodotus and Plato. Nonetheless, "with the growing emancipation of the individual from the old family solidarity", the former idea has to give way to the latter. And in turn, the notion of inherited responsibility that gradually becomes unacceptable prompts the maturation of hell by the introduction of new elements from eschatological movements. This paper is divided into five parts. The first part serves as an introduction. The second part discusses the Homeric depiction of the Hades, which represents an early Greek understanding of the life of the dead. The third part is devoted to a detailed analysis of Solon's notion of inherited responsibility and the various factors that contribute to its final explicit articulation. The fourth part focuses on the Orphic ideas of afterlife trial and transmigration of souls and their introduction into what we may call Platonic hell culminant in antiquity, which aims to offer a more self-contained system of justice and punishment. The fifth part is a conclusion.

**Keywords:** inherited responsibility; hell; divine retribution; justice; Solon; Plato; afterlife; transmigration of souls

## 1. Introduction

The problem of justice and punishment occupies a central place for the ancient Greek mind: will the evil receive its due punishment and the good be rewarded in proportion to its merits? The human empirical observation proves otherwise. As E. R. Dodds has shrewdly observed, "the Greeks were not so unrealistic as to hide from themselves the plain fact that the wicked flourished like a green bay-tree" (Dodds 1951, p. 33). The various ways to solve the difficulties intrinsic to this problem have witnessed the efforts of the Greeks in different historical phases. The urgency, sometimes greater and sometimes

less critical, prompted the ancient Greek authors to revise traditions and innovate new approaches in giving watertight replies to theological challenges. The related notions of hell, the afterlife, and the transmigration of souls are all products that aim to offer solutions to this demanding problem.[1] Other than these ideas that tend to ascribe the culpability to the transgressors themselves, there is one notable but long-ignored tradition that the punishment will be transmitted to the descendants of the sinners if they do not receive deserved retribution in their lifetime. Both traditions have co-existed in the Greek religions, and their competition for dominance in constructing a systematic theodicy runs through the history of Greek literature. Among the authors that are particularly relevant to our present concern, Homer, Hesiod, Solon, Theognis, Pindar, Herodotus, and Plato, who are deeply troubled by the problem of divine retribution and the establishment of a just cosmic order, are the main ones to be discussed in this paper.

Dodds believes that it was in the Archaic Age that the Greeks, prompted by the pervasive pessimism that "the mills of God ground so slowly that their movement was practically imperceptible save to the eye of faith" (Dodds 1951, p. 33), started to reflect on the issue of divine justice and retaliation. In order to sustain the belief that God supervises world order, it is inevitable and necessary to shake off the natural time-limit set by death, considering the hard experience that crimes usually go away without being revenged. If one looks beyond that limit, only two options are viable: "you could say that the successful sinner would be punished in his descendants, or you could say that he would pay his debt personally in another life" (Dodds 1951, p. 33).

Concerning the wane of the idea of inherited responsibility in classical antiquity, Kenneth Dover also offers an aptly historical observation. He suggests that "once civilized reflection had begun to suggest that it was incompatible with any notion of justice to hurt an individual because of something which that individual had not himself done, and at the same time the emotional need to believe in the punishment of wrong remained imperious, the notion of inherited punishment was gradually replaced by the notion that the soul of the offending individual himself would be punished after its separation from the body" (Dover 1974, pp. 261–62), and this leads us naturally to the constructive efforts of hell.

While previous scholars (E. R. Dodds and Kenneth Dover) have given conclusively the general outline, unfortunately only in passing, of how the notion of inherited responsibility and the idea of punishing hell have interacted and how the former tradition has been finally replaced, if partially, by the latter in archaic and classical Greece, there is yet not enough elaboration on the inner logic that eventually makes the partial replacement happen and the historical sources both traditions draw on for their justification. Such an elaboration is necessary since it not only offers sufficient textual evidence for the conclusion but also reveals the respective rationale behind these two traditions and, more importantly for our current concern, helps to shed light on the context of the construction of hell in the fifth century, especially in the earnest efforts of Plato. Additionally, as a sort of emendation for the theory held by Dodds and Dover, it is also pointed out in this paper that these two traditions are not incompatible as the theory seems to indicate, since both could co-exist in the same text or author, although normally with an emphasis of one over the other. This paper will further scholarly understanding of these two lines of traditions by offering detailed textual evidence on how they have co-existed and how the latter gradually has prevailed over the former. The second part discusses the nature of the Homeric underworld, that is, whether it is a place that could be properly understood in the sense of later punishing hell. It is shown in this part that the Homeric underworld, in general, is simply a negative mirror to the home of living, although there are indeed certain signs of punishment for the ones who have sinned against the Olympian gods. Nevertheless, the basic elements, such as its geography and Minos as the judge of the dead, pave the way for later (re)constructions of hell discussed in part four. The third part explores the concept of inherited responsibility represented by Solon, which is also a common religious belief in the late archaic age.[2] This part shows from new perspectives how the notion of inherited responsibility acquires a rationalized form with the aid of the concept of causality in the

new Ionian natural philosophy and of the legal thought of *lex talionis*. By these two concepts, it is emphasized that the wrongdoer must suffer commensurately. But due to the incongruousness of theory and reality that the wrongdoer normally leads a wealthy life and dies unpunished, the theology as we see in Solon has to appeal to family solidarity, which could be well attested in the Homeric epics in the form of collective punishment in curses. The tension between family solidarity and individual awareness, as is clearly shown in Theognis, opens up a space for authors desperately looking for ways to construct a just world. This naturally leads us to part four, which focuses on Plato's constructions of hell through multiple dialogues. In this part, I argue that Plato's representations of hell are by no means uniform, which is an important and necessary extension of the general notion held by Dodds and Dover. Specific texts and discussions are used to show how he proceeds from emphasizing the existence of hell to detailing it to cater to different philosophical needs.

## 2. Hades in the Homeric Epics: "Negative Mirror to the Home of Living"

Radcliffe Edmonds distinguishes three different levels of complexity in the ways that the ancient Greeks think with ideas of the afterlife, and the three categories might be continuation, compensation, and cosmology (Edmonds 2021, p. 12). This division provides a basic framework to understand the Greek hell. The continuative type of hell is as it sounds: those who have passed away are assumed to keep on living in the same fashion that they did in life, with the same basic identity and characteristics. Compensation involves imagining the dead doing the things they failed to do when alive but should have done then. It aims to offer a solution for the shared human experience that the unjust do not suffer punishments that are proportional to the crimes they committed, and the good do not receive adequate rewards for their actions in life. This type of afterlife imagined is closely tied to the individual's particular life, but instead of a simple reflection of that life continuing on in the afterlife, the compensatory afterlife provides "a negative image, reversing the wrongs of life and filling in the gaps" (Edmonds 2021, p. 12). The cosmological vision of the afterlife moves further than the second type. This type of afterlife is conceived as a complex system involving not only individual spirits but also the entire workings of the universe. Rather than examining the destiny of individual people, such concepts involve broader inquiries regarding the soul's entrance into the body at birth, the bond between soul and body during life, and the soul's course after death, when it is liberated from the body.

As early as Homer, the subject of death and the beyond has captured poetic imaginations. Hades in the Homeric epics in general is, to use Edmonds' classification, a continuous type of hell. It is merely a final destination where the souls convene when the living bodies are deprived of lives, or to quote Anthony Hooper, the Homeric Hades is "a realm that stands as a negative mirror to the home of the living" (Hooper 2021, p. 158), even though conflicting ideas are to be found within the same section.[3]

The most concentrated descriptions of Hades are found in the eleventh book of the *Odyssey*, where Odysseus takes Circe's advice and goes to Hades, the realm of the dead, to meet the prophet Tiresias. The scene is traditionally called *Nekyia*, a rite by which ghosts are called up and questioned about the future. It is normally accomplished with the *katabasis*, i.e., the physical journey to the Underworld. One digression must be made at this point concerning the common mistake of identifying Tartarus as Hades in the Homeric epics. In the scattered and scanty portrayals of Tartarus in the Homeric epics, it is murky yawning under the ground, "far, far away, where is the deepest gulf beneath the earth, the gates whereof are of iron and the threshold of bronze, as far beneath Hades as heaven is above earth" (*Iliad*, 8.13–16). The reason for the common mistake is evident in that both share the same characteristic, which is darkness. The primary feature of Hades (in Greek Ἀΐδης, whose literal meaning is "unseen") as the abode of the dead is the absence of light, in contrast with the world of the living. While our world is one lived in the presence of sunlight, Tartarus, as the hell for the divinities who sinned against the gods (Cronos, for instance), is shut off even against the penetrating rays of Helios:

The undermost limits

Of earth and sea, where Iapetos and Kronos seated

Have no shining of the sun god Hyperion to delight them

Nor winds' delight, but Tartaros stands deeply about them. (*Iliad*, 8.478–481)[4]

The *Odyssey* offers a vague guide to arriving at the entrance of Hades. To reach there, a living person must sail West of the fairy-lands, far out to sea (*Odyssey*, 11.13–16). Nevertheless, of the landscape of the abode of the dead itself and its furnishings, the poet tells us nothing. The sacrificial ceremonies to prepare Odysseus to enter Hades receive a detailed and dramatic description. He pours full of drink offerings for all the dead, first honey mixed with milk, the second pouring sweet wine, and the third, water, and over it all, he sprinkles white barley. Then, he took the sheep and cut their throats over the pit, and with the dark-clouding blood running in, "the souls of the perished dead gathered to the place, up out of Erebos, brides, and young unmarried men, and long-suffering elders, virgins, tender and with the sorrows of young hearts upon them, and many fighting men killed in battle, stabbed with brazen spears, still carrying their bloody armor upon them" (*Odyssey*, 11.35–41). Obviously, the dead souls in hell retained their life-time physical features, but their form of infernal existence changed drastically. In the episode where Odysseus encounters his dead mother, Anticlea, when he attempts to hold her in his arms, he finds that she eludes his embrace.

It was fain to clasp the spirit of my dead mother. Thrice I sprang towards her, and my heart bade me clasp her, and thrice she flitted from my arms like a shadow or a dream. (*Odyssey*, 11.205–207)

This is not the place to further the discussion of the thorny subject of "spirit" (or "soul" and many other translations of the Greek word *psuke*). Generally speaking, in the characterization of the inhabitants of Hades, words such as *phrenes*, *noos*, and *thymos*, among several others, are used to describe the dead, in contrast to the world of the living. The words *skiai* ("shadows," "shades") and *eidolon* ("image" or "phantom") are the most common ones to describe the shape and appearance of those who lack true substance caused by death. Regardless of the different terms, one prominent feature of the dead souls in Hades is their lack of corporeality, which, to a large extent, reflects an idea that is in constant tension with the concept of hierarchical distinction and individual destinies that we later see in the punishing hell. What is implicit in both of these episodes, which depict the helplessness of the living as they watch their loved ones slip away from them, no matter how hard they try to hold them in their arms, is the fact that death is an inevitable part of life and it will never be possible for anyone to avoid such a pathetic existence. Penelope explains to her dear son why this is a common law for every mortal: "the sinews no longer hold the flesh and the bones together, the queens of the past and once the spirit has left the white bones, all the rest of the body is made subject to the fire's strong fury, but the soul flitters out like a dream and flies away" (*Odyssey*, 11.219–222).

In other words, the decomposition of the body, which is hastened by cremation, marks the end of life. However, what remains in the realm of the dead appears to retain its physical form if one views it as capable of performing duties or being punished. Therefore, from the point of view of the living, the deceased are mere shadows or ghosts in the afterlife, devoid of any sign of life, yet, despite being disembodied, they retain their physical form. Such a dreary vision of an afterlife with no substance, joy, or feeling is actually in line with the social function of the poetic genre epic, which is deemed by the Homeric rhapsodes to be the only means to provide any kind of meaningful immortality or real afterlife (Edmonds 2013, pp. 264–67). Overall, the Homeric epics do not attempt to portray Hades as a place of retribution or punishment for a life lacking in virtue. Retribution, as far as we can tell from existing epic materials, is not the primary purpose of Hades. It may be possible for the Homeric audience that Hades is viewed as a punishment for all the dead in view of their pitiable and wretched existence.

But a few outstanding exceptions make the Homeric vision of Hades complicated, namely the famous cases of Tantalus, Tityus, and Sisyphos. Homer departs from the usual narrative of hell with these three figures, who suffer for their sins against the gods and the cosmic order. Sourvinou-Inwood, for instance, observes that outside those deep-rooted ideas about destiny beyond death, there are indications that "a new eschatology has begun to supplant the archaic tradition by which the psyche and its residual vitality were understood" (Sourvinou-Inwood 1995, p. 72; Mirto 2012, p. 18). Beliefs that the fate of the dead is not shared by all are beginning to surface, though this newfound perspective only applies to a select few. In the *Odyssey*, for instance, the sea god Proteus prophesies that Menalaus' fate would be radically different from that of his companions Ajax, Agamemnon, and Odysseus.

> You shall die and go to your end in horse-pasturing Argos, but the immortals will convoy you to the Elysian Field, and the limits of the earth, where fair-haired Rhadamanthys is, and where there is made the easiest life for mortals, for there is no snow, nor much winter there, nor is there ever rain, but always the stream of the Ocean sends up breezes of the West Wind blowing briskly for the refreshment of mortals. (*Odyssey*, 4.561–69)

This is one of the earliest literary evidences of the Isle of the Blessed, to which the good people are sent as a reward for their life-time virtue.[5] Likewise, Hades as a place for post-mortem punishment is also occasionally seen in the Homeric epics. The exceptions constitute the cases of Tityus (11.576–81), Tantalus (11.582–92), and Sisyphus (11.593–600). The gods alone have decreed for them an incomparably harsh fate to continue their suffering throughout eternity, making it clear that their punishment is special and not part of a universal pattern of post-mortem retribution. The myths concerning these three mythical characters are not seen in Homer, but later traditions refer indubitably to the nature of their crimes. Tityus attempts to rape Leto (Tityus' name is probably derived from the Greek word $\tau\acute{\iota}\sigma\iota\varsigma$ meaning "he who suffers retribution"); Tantalus offers up his son, Pelops, as a sacrifice by cutting him up, boiling him, and serving him up in a banquet for several gods in order to test their omniscience; Sisyphus chains Death when he comes to fetch him. Their punishments in the Homeric Hades are gruesomely detailed.

> And I saw Tityos, son of glorious Gaea, lying on the ground. Over nine roods he stretched, and two vultures sat, one on either side, and tore his liver, plunging their beaks into his bowels, nor could he beat them off with his hands. For he had offered violence to Leto, the glorious wife of Zeus, as she went toward Pytho through Panopeus with its lovely lawns. (*Odyssey*, 11.576–81)

As a result of his offense, Tantalus is placed in a pool of water beneath a fruit tree with drooping limbs. Every time he attempts to pluck the fruit, the tree's branches would deny him of the delicacy. Every time he leaned forward to take a drink, the water would vanish before he could quench his thirst. The revenge for Sisyphus is much more well-known: he endlessly toils, perpetually pushing a massive boulder up the incline, only to roll back down the second it nears the summit. Homer departs from the typical afterlife narrative with these three figures, who suffer for their sins against the gods and the cosmic order. However, on the more general level, there is no solid evidence of a universal judicial system of post-mortem punishment in the Homeric Hades.

The inconsistency caused by these exceptions invites various interpretations, which could be roughly categorized into two groups. One group tends to believe that the belief of post-mortem punishment already existed in Homer's time, but the poet's agenda makes him expunge some of them. Alfred Heubeck is a representative of such an interpretation by concluding on a firm basis that "by contrast with the diversity of current views of the soul, the underworld, and life after death, the poet, in general, presents a conception in which very various elements are unified into a single coherent picture; but, from time to time we catch a glimpse of popular beliefs which he has otherwise excluded" (Heubeck and Hoekstra 1990, p. 112). The other group of scholars insists that the verses of 11.568–627

are an interpolation of a much later tradition, notably the Orphic beliefs of hell and the afterlife, which could be traced back to the 6th century BC. The problem with the former interpretation that insists on the existence of post-mortem punishment lies in the lack of internal evidence from Homer, in which the notion of hell as a place for afterlife trial has only a dim presence.

The most direct support for this conclusion is that Hades is not considered a god who judges the dead for their actions while alive, as he is expected to be. Although such an idea does make its appearance occasionally in poetic imagery, particularly when he is depicted as the chthonic counterpart of Zeus, the supreme god of justice (Aeschylus, *Suppliants*, 228–31l; *Eumenides*, 273–75), his usual attitude is "absolute neutrality" in the Homeric epics (Mirto 2012, p. 22). The passage that precedes the descriptions of the three divinities who transgress the cosmic order may easily confuse our general understanding of the Homeric hell.

> There then I saw Minos, the glorious son of Zeus, golden sceptre in hand, giving judgment to the dead from his seat, while they sat and stood about the king through the wide-gated house of Hades, and asked of him judgment. (*Odyssey*, 11.568–571)

The tradition of Minos as the infernal judge is nowhere else found in Homer. As multiple scholars have pointed out, "it was only in later tradition that Minos became the judge of the dead," most notably in Plato's *Gorgias*, 523e6–7 (Heubeck and Hoekstra 1990, p. 111). Furthermore, it is apparent from this passage that Minos does not judge the souls in order to decide their fate in the Underworld since nearly all the souls in the Underworld are trapped in miserable conditions. The task of Minos at most lies in "administering justice and resolving conflicts between the deceased in Hades" (Bernabé 2021, p. 139), and this reading is susceptible to the critique of stretching a point.

To conclude this part, the Homeric account of the Underworld represents the standard early Greek view of the afterlife in Hades, which is inhabited by the pale shades and miserable phantoms of the perished dead. Although there are signs that are read as suggestions of "the compensatory afterlives that rectify the incompleteness of justice in the mortal world" and "the grand cosmic visions that bring together life and death, mortal and immortal, chthonic and celestial" (Edmonds 2021, p. 12), the Homeric Hades is, in general, a dwelling house for the deceased where there is no obvious indication of a hierarchical treatment for them. A fragment of Sappho well represents such a living condition. The spirits of the deceased flutter anonymously and aimlessly, like bats in a cave or shadows on a wall, rather than remaining as souls that carry on in the Underworld the lives they did in life.

> Dead you will lie, nor will there ever be any memory of you, not at any time later, for you have no share in the roses of Pieria. But unnoticed even in the halls of Hades, you will wander, flitting among the dim dead. (Sappho fr. 55)

The Homeric depiction of Hades is not particularly concerned about cosmic justice and punishment, with only a few exceptions, such as those who are tortured for their hubris against gods. A more significant innovation of Hades occurs with the urgent quest for a reasonable explanation for human transgression and suffering. This will be discussed to a greater extent in Section 4. Before continuing our investigation of hell in Greek antiquity, our next focus will be the neglected notion of inherited responsibility, a parallel idea with the imagination of a punishing hell.

## 3. Inherited Responsibility in Solon, Theognis and Herodotus

The concept of inherited responsibility in religion, which is explored in ancient and classical Greek literature, has garnered significant attention from scholars. Following Gustave Glotz's groundbreaking work *La solidarité de la famille dans le droit criminel en Grèce* (1904), which provides a comprehensive and innovative treatment of inherited responsibility, E. R. Dodds in his book *The Greeks and the Irrational*, examines the concept within

the context of Greek civilization's development and presents it as a significant phase in the evolution of the Greek mindset. According to Dodds, the concept of the archaic age's characteristic doctrine of belief was only one of the two possibilities for people of that age, when "the mills of God seemed to turn so slowly that their motion was almost unnoticeable, unless you had faith," and "in order to sustain the belief that they moved at all, it was necessary to get rid of the natural time-limit set by death" (Dodds 1951, p. 33).[6] The other, understandably, points to the idea that the sinner would pay his debt personally in another life. Dodds' interpretations of it were later largely modified by Hugh Lloyd-Jones in *The Justice of Zeus*. Unlike Dodd's interpretation, which views the notion as an unjust and superstitious belief stemming from untamed irrational impulses, Lloyd-Jones sees it as a fundamental aspect of the Greek understanding of cosmic justice and a natural mechanism (Lloyd-Jones 1983, p. 107).

The Athenian statesman and lawmaker Solon represents a different yet equally influential tradition from that of Hades in the Greek archaic age. The bold articulation of inherited responsibility implies a denial, at least a partial one, or an intentional omission of the very existence of Hades.[7] The absence of afterlife punishment compels Solon to reconstruct his own system of justice and world order. One needs to note that the idea of inherited responsibility has made its appearance in the Homeric and Hesiodic epics, not as a punitive mechanism, but in the form of the threat of collective punishment in oath-swearing instead of being a rational construction of theodicy in Solon. To a certain extent, the bold articulation of inherited responsibility in Solon is heavily based on the popular notion of family solidarity, which could be testified in Homer and Hesiod.

In the *Iliad* Book II, the Trojans march from the city gates and advance to meet the Achaeans. Paris, the Trojan prince who initiates the war by stealing the beautiful Helen from her husband, Menelaus, challenges the best of the Achaeans to fight with him face-to-face in dread combat (2.20). When Menelaus, dear to Ares, steps forward, however, Paris loses heart and shrinks back into the Trojan ranks. Hector, Paris' brother and the leader of the Trojan forces, mocks the cowardice of Paris that "there is no strength nor valor in his hear" (45) and readdresses the issue of the cause of this nine-year war, its disastrous consequence, and the tragic involvement of his father, his city and his people. Deeply humiliated by Hector's insult, Paris finally agrees to a duel with Menelaus, declaring that the contest will establish a tentative peace treaty between the Trojans and the Achaeans by deciding once and for all that whoever wins the battle shall have Helen as one's wife, while other people who are outside the battle should remain friendly to each other. Hectors says this and calls for a sacrifice in attestation of a trustworthy oath (95). Menelaus, however, insists that Priam is brought in order that "he himself cut the oath with sacrifice" (105–106), taking into consideration that his sons are "arrogant and unfaithful" (106).[8] Thus, we arrive at "the great oath of the armies described in Books III and IV of the *Iliad*," "the earliest explicit attestation of ancestral fault in the record of Greek poetry" identified by Renaud Gagné (Gagné 2013, p. 363):

> "Zeus, most glorious, most great, and other immortal gods, which host soever of the twain shall be first to work harm in defiance of the oaths, may their brains be thus poured forth upon the ground even as this wine, theirs and their children's; and may their wives be made slaves to others". (*Iliad*, III. 298–301)[9]

The Homeric tradition of swearing an oath with the lives of family members is inherited by Hesiod when justifying the authority of Zeus in supervising justice. For him, whoever swears a false oath or tells a lie in his testimony harms justice. The horrifying consequence once again points to the collective fates of a household that the family of the perjurers will be left more obscure, whereas the family of the man who keeps his oath will be prosperous in the after times (*Works and Days*, 282–85). The idea of collective punishment and inherited responsibility is a strong and necessary guarantee of overarching justice in the absence of a punishing hell that exercises its judicial function on men. Solon is fully aware of this tradition. The essential difference between Homer's collective punishment and Solon's inherited responsibility lies in the perspectives. For the former, the collective

punishment serves only as a threat in the process of oath-swearing. It does not have any religious efficacy or theological support. The latter, however, is a theodicean construction, or rather a theological one. It aims to offer a consistent and meaningful justification for inherited responsibility and its role in maintaining justice in the world. Different from its representation in the Homeric and Hesiodic epics, he tries to shake off the capricious nature of divine justice and ascribe it to a rational explanation. In the most well-known fragment, 13, Solon utters for the first time in our known history of Greek literature, the principle of inherited responsibility. The transmission of retribution through the divine will as the foundation of cosmic justice is fully elaborated and rationalized into a self-contained system that is closely associated with the social and political ideal of balance.

It is unlikely that the idea of hell, or at least the Homeric idea of Hades, remains unknown to the poet since Homer's authority as the crown of poetic invention lasted throughout antiquity. The only deduction we are able to make at the moment to account for its absence is that various religions of eschatology have not yet transformed Hades into a place specifically designed for the afterlife judgment. The following section will be devoted to uncovering the way the notion of inherited responsibility comes to stand on its own, namely the conditions for its emergence, its inner logic, and most importantly, its basic difficulty that inevitably leads to its final replacement by the idea of hell, notably in Plato's construction of a just world. The key passage of our concern in Solon writes:

> One man pays the penalty at once, another later, and if they themselves escape the penalty and the pursuing destiny of the gods does not overtake them, it assuredly comes at another time; the innocent pay the penalty, either their children or a later progeny. (13.29–32)[10]

The temporal limit of an evildoer paying for his transgressions becomes prominently no longer a worry in this framework. The statement clearly brings forth the way divine justice exercises its forces. In the first place, punishment is inevitable; it either befalls the offenders or their offspring. Irrational as it may sound, the principle, in effect, has undergone a significant procedure of rationalization. Or rather, in view of the fact that the notion comes a long way before its final crystallization, the rational movement of the Ionian pre-Socratics contributes significantly to its expression. The justification of inherited responsibility starts with a revision of the divine nature. Compared with the inconstant nature of the divinities in Homer and Hesiod, the Solonian Zeus is also unpredictable, but in a distinctive way that he sees everything and waits until the right moment to act.

> He is not, like a mortal man, quick to anger at every incident, but anyone who has a sinful heart never escapes his notice and in the end he is assuredly revealed. But one man pays the penalty at once, another later, and if they themselves escape the penalty and the pursuing destiny of the gods does not overtake them, it assuredly comes at another time; the innocent pay the penalty, either their children or a later progeny. (13.25–32)

In another long fragment, 4, the poet also attributes Zeus' qualities to the goddess Dike, who acts as his agent. She remains silent while watching over the sinners as well (4.15–16). In the marked distinction, Solon draws between the majesty of divine intervention and the uncontrolled human anger, the problem of how god exercises his punitive powers of justice comes into view. It is, in effect, a response to the universal complaint that the sinners always escape their deserved penalty, and the good people generally suffer. In a theological framework that excludes hell, afterlife, or transmigration, Solon calls for people's patience and reveals the cryptic way in which god works: if the sinners do not receive their immediate penalty before their death, the retribution will befall their progeny.

At first glance, this deceptively blunt notion may easily be understood as merely an expression borrowed from popular imagination, as we have mentioned earlier, in the form of collective punishment. However, much innovation is attached to it through Solon's revision. For instance, its inherent relation to Hesiodic theology and Solon's ideal of civic and economic equilibrium, which is beyond the scope of this paper, reveals the tensions

between his use of mythical traditions and the more urgent present. In this paper, its efficacy as a system that aims to punish the evildoer shall be our central concern, and its accomplishment, somewhat ironically, is made possible by the rational movement of the so-called Ionian philosophy in the 6th century BC, notably Anaximander.[11] Solon certainly witnesses the transition from the mythical interpretation of the universe to the rational explanation of the cosmological order. More specifically, Anaximander's notion of negative reciprocity (τίσις) greatly shapes Solon's idea of divine retribution through descent.

Quite different from his predecessors like Hesiod, Solon makes it explicit that no change is arbitrary, and whenever a change happens, there must have been a disturbing cause. By way of causality, the notion of inheriting ancestral sins acquires a solid theoretical framework that makes the retribution feasible within the confines of families. The divine retribution on communities through the capricious will of gods now becomes a particular punitive mechanism that traces the source of the sins and places a burden on the descendants of the evildoers. The punitive mechanism of transmitting punishments to one's progeny originates partially from the proverbial notion of *lex talionis* through the terminology τίσις, which indicates the act of "payment by way of return" or "recompense and retribution." τίσις, in its most basic sense, denotes transactional retribution. The early evidence of legal documents, the Gortyn law code, for instance, attests to its connotation of the reciprocal exchange of one harm for another, normally in the form of economic exchange. The *lex talionis* requires the immediate return of an identical harm, or to quote Alexander Loney, "avengers feel retribution is at its purest when a wrongdoer simultaneously harms himself by his own attack" (Loney 2019, p. 28). The talionic principle that emphasizes one's own responsibility for the offense is not rare in the texts both before and after Solon's age. This notion of a single act serving as both offense and punishment could be traced to as early as Hesiod. "If someone sowed evils, he would reap evil profits; if he suffered what he committed, the judgment would be straight" (Hesiod, *Fragments*, 286).[12] The idea of "sowing" evil that the offender will reap later could be understood on two levels: (i) the offender incurs punishment later in his life instead of a harm that occurs instantaneously; (ii) the offender leaves his offense to be atoned by the descendants. Both interpretations point to the atemporal and diachronic aspects of negative reciprocity, even though they are not distinctively distinguished from each other until then. Another level of meaning of this quote may point to the compensation of the offender himself, and this interpretation is natural given the later introduction of the punishing hell, in which the evildoer pays for his own wrongdoing, even though the idea of hell is nowhere to be found in Hesiod. The central argument lies in the temporality of retribution.

In the context of divine justice, its punitive activities, unlike the *lex talionis* as a human law, often postpone themselves, as one commonly experiences in reality. The temporality of negative reciprocity also baffles Solon's contemporaries, and a striking parallel of Solon (13.25–35) is found in Theognis.

> Whatever possession comes to a man from Zeus and is obtained with justice and without stain, is forever lasting. But if a man acquires it unjustly, inopportunely, and with a greedy heart or seizes it wrongly by a false oath, for the moment he thinks he's winning profit, but in the end it turns out badly and the will of the gods prevails. The minds of men, however, are misled, since the blessed gods do not punish sin at the time of the very act, but one man pays his evil debt himself and doesn't cause doom to hang over his dear progeny later, while another is not overtaken by justice; before that ruthless death settles on his eyelids, bringing doom. (Theognis, 197–208)[13]

The context of the discussion on the receivers of the atonement points to the just and unjust ways of acquiring wealth, in view of which one may deduce a shared concern on the issue about the persons who are supposed to be the targets of divine vengeance. Theognis' stance is ambivalent and indeed has a strong sense of hesitance. The absence of elements such as hell or transmigration in Theognis, which Solon surely shares, further complicates the matter. The Theognidean anxiety lies in the law's delay (193–194) and the natural life

limit of the offending agent (207–208). In his view, the injustice is definitely to be atoned. One pays for the offense by himself; another dodges the hunting of divine retribution since death terminates his life before justice is served. It is only in the negative affirmation that if the former case happens, the offspring of the offender will not have to take on the burden of their ancestors that Theognis responds to the basic difficulty of justice in the theological sense. Later in the Theognidean corpus, not surprisingly, one reads that the poet intensely opposes the transmission of sins to children who are faithful to Zeus.

> Whosoever did acts abominable and of intent, disdainfully, without regard for the gods, should then pay woeful requital in person, and the father's sins should no longer be a bane for his sons afterwards; and would that sons of an unjust father who act with just intent, dreading your anger, son of Cronos, and loving justice from the start in their dealings with fellow townsmen, should not pay for the transgressions of their fathers. (Theognis, 734–740)

The predicament is not easy, especially when the lives of descendants as private men are not viewed as merely nonessential constituents of a community. On the one hand, Theognis announces the principle that the offender should make atonement for oneself (736); on the other, the careful subjunctive mood Theognis deploys to entreat Zeus not to make it befall the descendants may reflect a personal struggle against a general trending belief. The irreconcilable gap recognized by Theognis between the human law and the divine law to which the former ultimately has to appeal on account of its natural limit questions for the first time the Hesiodic assumption that the communal life benefits and suffers as a whole. Solon must have realized it as well, by claiming that even those children who are "innocent" (13.31–32: ἀναίτιοι ἔργα τίνουσιν ἢ παῖδες τούτων ἢ γένος ἐξοπίσω) are doomed to take on the guilt. The word ἀναίτιοι ("innocent") is nowhere else found in Hesiod, even in Theognis.[14] Despite that, one needs to note that Solon is primarily dealing with the way Zeus exercises his divine power rather than trying to offer a practical judicial system, although the basic formula in constructing the theory of cosmic order is heavily reliant on the existing institution of solving disputes. One valuable fragment of Anaximander consisting merely of a single sentence will provide a significant background for understanding Solon's statement on inherited sins and its innate connection to the theoretical construction of cosmic order. It reads as follows:

> ἐξ ὧν δὲ ἡ γένεσίς ἐστι τοῖς οὖσι, καὶ τὴν φθορὰν εἰς ταῦτα γίνεσθαι κατὰ τὸ χρεών· διδόναι γὰρ αὐτὰ δίκην καὶ τίσιν ἀλλήλοις τῆς ἀδικίας κατὰ τὴν τοῦ χρόνου τάξιν.

> But from whatever things is the genesis of the things that are, into these they must pass away according to necessity; for they must pay the penalty and make atonement to one another for their injustice according to Time's decree.[15]

The similarity between Solon 13.29–32 that makes the statement of inherited responsibility and the words quoted above is obvious (ἀλλ᾽ ὁ μὲν αὐτίκ᾽ **ἔτεισεν**, ὁ δ᾽ ὕστερον— **διδόναι γὰρ αὐτὰ δίκην** καὶ **τίσιν** ἀλλήλοις τῆς ἀδικίας; 4.16: **τῷ δὲ χρόνῳ** πάντως ἦλθ᾽ ἀποτεισομένη—**κατὰ τὴν τοῦ χρόνου τάξιν**). Werner Jaeger's groundbreaking observation is worthy of our citation at this point. He believes that "Solon's new and deep experience of the divine springs from his intuition of a meaningful immanent law ruling the social life of man and automatically keeping the balance of justice" and that this immanent law is analogous to the speculation taught by the Milesian natural philosopher Anaximander a decade later (Jaeger 1966, p. 92). The only Anaximandean fragment preserved invites various interpretations (Jaeger 1947, p. 34, n. 53), but the general image of a scene in a courtroom receives unanimous agreement among most scholars. The reciprocal principle explicit in this image is once again related to settling property. As Jaeger suggests, the image denotes a dispute involving two parties, in which the one who has taken more than his share must pay damages for his pleonexy to the party he has wronged (Jaeger 1966, p. 94).

Up until this moment, the principle itself is sufficiently clear. But to further complicate the matter, one is tempted to ask the question: if the person who enjoys the benefits of having too much has to make atonement, it should be understandably reasonable that the due punishment should occur instantaneously once the judicial decision is made; in view of this, what is the point of concluding the statement with κατὰ τὴν τοῦ χρόνου τάξιν ("according to Time's decree")? Suppose one determines to take the temporality of retribution seriously into consideration. In that case, only two viable options are available in light of Jaeger's efforts in connecting "the world of politics" with "the whole realm of Being": either Anaximander demonstrates the constant interchange of cosmological opposites into primal equilibrium, as Simplicius and Kirk have suggested; alternatively, the statement of "Time's decree" must be interpreted in the way that the cosmos, with nature and human society involved, may seem unjust at the moment, but this is only a temporary arrangement since the retributive force that works itself out diachronically will eventually restore the primordial equilibrium. It is certainly easy for the cosmological interpretation that tosses out its concerns for an individual human and detaches itself from human affairs to acquire a justification. However, when applied to the political world, the retributive mechanism of the cosmos meets the basic difficulty that we have already mentioned: the immediacy required by retribution in the legal sense.

If one insists on the consistency between the divine law and the human law, one final converging point is inevitable, and Solon is the first Greek author to announce it by the statement, although already partially anticipated in Hesiod, that the divine retribution will either be inflicted on the sinner himself later in his life, or transmitted to his progeny. Its complete and confident articulation takes nearly a century's preparation, notably under the influence of the Ionian philosophy: the theory of cause and effect, the reciprocal principle of cosmic order, and the abstraction of time. Although the word τάξις never appears in the extant fragments of Solon and could not be attested in Hesiod, it is clear that in Solon, the cognitive force attributed to Dike with her only weapon "time" eventually makes the whole punitive procedure self-justified.[16] In other words, the Solonian justice through the mechanism of inherited responsibility has to gain the support of time. If one sinner escapes, time will be the weapon that tracks him/her down.

> They (i.e., the rapacious wealth-stealers) have no regard for the august foundations of Justice, who bears silent witness to the present and the past and who in time assuredly come to exact retribution. (4.14–26)

Based on this passage and multiple others, Emily Anhalt correctly observes that Muses' *mnemosyne* (13.1) and Dike's "silent witness to the present and the past" (4.15) "as the knowledge of cause and effect is necessary to harmonious political organization since this knowledge is essential to a sensible guide to conduct" (Anhalt 1993, pp. 19, 68). The causal connection between a specific human action and its consequences guaranteed by divine forces helps a person to deduce that the infatuation that befalls a person must be traceable to a certain more or less distant familial source. Readers may have fundamental doubts regarding this punitive mechanism. On the one hand, if an offender escapes his punishment and lives a well-off life, the whole foundation of justice risks itself being completely overthrown; on the other, if the offender's descendants, innocent and living a just life, fall heir to the due revenge, it is hardly sufficiently legitimate to call it justice. Furthermore, if one shifts the perspective, things become much more complicated. Think of a person who acts justly but fares ill. Relying on this theory, he may be well tempted to ascribe his misfortune to an ancestral fault; consequently, its inevitable product is fatalism. In effect, Theognis' poetic sensibility has captured the dubious aspects of inherited responsibility.

> But now the perpetrator escapes and another then suffers misery. Also, king of immortals, how is it right that a man who keeps from unjust deeds and does not commit transgressions and perjury, but is just, suffers unjustly? What other mortal, looking upon him, would then be in awe of the immortals? What frame of mind would he have whenever an unjust and wicked man who does not avoid

the wrath of any man or god commits wanton outrage and rolls in wealth, while
the just are worn out and consumed by harsh poverty. (Theognis, 743–752)

While continuing his earlier entreaties that the guiltless should not be burdened with
divine retribution, Theognis turns unexpectedly to the present tense as if making a factual
statement. The factual statement is followed by a long list of questions that are skeptical of
the validity of the justice that punishes the guiltless ones. It certainly makes sense to treat
Theognis' sharp series of questions along the lines of Solon and even so to consider them as
a protest against Solon's bold articulation or the articulation in a much broader context as
the recapitulation of a popular religious belief. And with the rise of the sense of individual
responsibilities as Theognis represents, the notion of inherited responsibility fades away
in popular imagination and literature and is gradually superseded by the Orphic notion
of afterlife judgment, which finally gets incorporated into the Platonic philosophy.

The Solonian notion of inherited responsibility was taken over by Herodotus, who
was active in the 5th century, and it serves as a piece of good evidence for its influence at
this age. Unlike Solon, in Herodotus, the awareness of both traditions of inherited respon-
sibility and of Hades is traceable, even though the former plays a much more influential
role than the latter in the narrative. Let us discuss Herodotus' appropriation of Solon's idea
of inherited responsibility in the first place. It has been shrewdly observed by Ryan Balot
that "the classical discourse on greed, which is initiated by Herodotus, borrows language,
concepts, and themes from Solon's attempt to articulate and resolve the problems posed by
unjust acquisitiveness within the political community of archaic Athens" (Balot 2001, p. 99).
Also, by no means is it a coincidence that Solon is the first Athenian to enter the world of
the *Histories* and to act as an Athenian sage in advising the tyrant Croesus on the subject
of the relation between the and happiness, which is a classic Greek theme already both
in the archaic and classical age.[17] The first contact between the Greeks and the barbarians
starts with Solon's voyage (θεωρίη) to Sardis, and more importantly, Solon's speech to the
acquisitive Croesus becomes the Herodotean pattern for later historical developments re-
volving around mutability of human fortune, divine jealousy, and the boundaries between
the two worlds.

Herodostus' representation of inherited responsibility is implicit in his characteriza-
tion of Solon and reconstructions of Solon's doctrines. Solon's visit covers almost ninety
chapters (1.6–94) in the opening of the *Histories*. The Croesus Logos consists primarily of
three parts: the Lydian accounts of the conversation with Solon (1.29–33), the tragic death
of Croesus' son Atys (1.34–45), and the fall of Croesus (1.85–89). In the account of Solon's
visit to Lydia, Croesus' pride on account of the vast amount of wealth and empire he owns
and his disdain for Solon's esoteric teaching about the variable nature of human affairs and
the way gods manipulate their powers on injustice (1.32) eventually fails him in giving in-
sight into the imminent misfortunes and their causes. Solon shows that among men who
are "entirely subject to chance" (1.32.4) and happen to enjoy a lifetime of seven decades,
only the one who "he chances to end his life with all well" (1.32.5) could be considered
happy.[18] Wealth is not the condition for happiness. Solon continues his Herodotean dialec-
tics. The wealthy but unfortunate man surpasses the lucky one in only two fields, whereas
the fortunate one surpasses the affluent but unfortunate in many. The wealthy person
is more capable of gratifying his desires and sustaining a massive tragedy that comes to
him, and it is in these areas that he surpasses the other. Though the fortunate man may
not be as capable of withstanding disaster or hunger as the affluent man, his luck protects
him from them, and he is free from deformity and illness, has no experience of evils, and
has attractive children and a pleasing appearance. What is implied in Solon's ambiguous
teachings is that the continuation of a family's lineage is much more important than the ac-
quisition of wealth. Additionally, true happiness lies in things that end well, which clearly
indicates how he finally becomes the victim of his ancestor's transgressions when the real
cause of familial destruction is disclosed towards the end of his life. The teachings of the
Herodotean Solon about the unknowability of the divine will and men's fortunes affected
by it suit fragment 13 well. However, the central notion of inherited responsibility is not

explicitly presented in Solon's speech. It is only alluded to through verbal implications of "uprooting" (1.32.9) and the thought that children are deemed as an essential reference in judging if a person is truly entitled to the label of happiness.[19] In spite of that, the fact that Herodotus interweaves the Solonian notion into the fall of the Lydian empire leads the doctrine in a new direction. Instead of being an earnest doctrinal effort to establish the system of justice as in Solon, the principle of inherited responsibility becomes a dynamic narrative strategy that points up the sinister consequences inflicted on the barbarian dictators with extraordinary evil deeds in Herodotus and acquires a new connection with the Delphic oracle. The historical mission of representing the Greek ideology and initiating the whole paradigmatic principle falls understandably upon Solon.

After a retrospective account of the causes of the Trojan War between Greece and the East, the historian introduces Croesus, "the first man whom I myself know began to commit unjust deeds against the Greeks (1.5.3)". The historian traces the history to the last kind of Lydia's Heraclid dynasty, which was succeeded by the Mermnad dynasty founded by Gyges, instead of embarking immediately on a narrative about Croesus. The dramatic narration on the origin of the Mermnad dynasty, also the source of a series of later afflictions, is an episode of a private moment in the court, filled with unexplained, perhaps unexplainable, thoughts. Under the compulsion and contrivance of Candaules, Gyges, the king's most trusted bodyguard, hid in Candaules' bedroom and, when the queen entered, watched her undress. The queen, having been shamed, swore secretly to avenge herself. The day after, she commanded Gyges to her room. Gyges thought it would be a common request, yet he was surprised when she instantly confronted him and gave him two alternatives: the one was to murder Candaules and ascend to the throne with Nyssia as his spouse; the other was to be killed instantly by her dependable servants. Gyges eventually decided to take the first course of action and assassinate the king. With the throne in hand, in order to come to an agreement with the rebellious Lydians, Gyges consulted the Delphic oracle on condition that if the oracle should ordain him, king of the Lydians, then he would reign; but if not, then he would return the kingship to the Heraclidae. The oracle ordained, but the Pythian priestess declared that the Heraclidae would have vengeance on Gyges' posterity in the fifth generation (1.13.2). It is emphasized by Herodotus that the Lydians and their kings paid no regard before it is fulfilled (1.32.3). The kingly lineage of the Mermnad dynasty was clearly outlined by Herodotus, even though the voice uttering the inherited responsibility of Croesus is not to be heard once again until the king's final moment: Gyges ruled Lydia for thirty-eight years (1.13); his son Ardys reigned for forty-nine years and was succeeded by his son Sadyattes, who reigned for twelve years (1.14–15); and after Sadyattes came Alyattes, whose reign lasted fifty-seven years (1.25); Croesus came to the throne and became the fifth generation since Gyges.

The Solonian notion of men's failure to mark the way things end is repeatedly characterized as a personal flaw of Croesus, specifically his heedless (mis)interpretation of the Delphic oracles, even though he cultivated whole-heartedly the good relations between Lydia and the sanctuary of the god Apollo in Delphi on continental Greece first established by his great-great-grandfather Gyges and maintained by his father Alyattes (Gyges: 1.14; Alyattes: 1.25; Croesus: 1.46–56). Two cases stand out. Threatened by the rise of the Persian empire under the rule of Cyrus, Croesus sent delegates to Delphi to inquire if he should command an army against the Persians. The oracle prophesied that "if he should send an army against the Persians, he would destroy a great empire" (μεγάλην ἀρχὴν μιν καταλύσειν). Exalted by the positive reply, Croesus offered the sanctuary rich presents in dedication and made a third inquiry of the oracle concerning whether his sovereignty would be of long duration. The Pythian priestess answered:

When the Medes have a mule as king,

Just then, tender-footed Lydian, by the stone-strewn Hermus

Flee and do not stay, and do not be ashamed to be a coward. (*Histories*, 1.55.2)

The result of Croesus' misapprehension of the Delphic oracles is catastrophic.[20] In the dramatic scenes invented by Herodotus of the fall of Sardis, Croesus, who had been saved from being burned on the pyre by the Persian Cyrus, requested permission from his victorious opponent to send the chains to the god of the Greeks and a delegation to Delphi to ask Apollo for explanations on "his way to deceive those who serve him well" (1.90.2). He wished to "reproach the god for this (capture)" (1.90.3) and to ask the god if he were not ashamed to have persuaded Croesus to attack the Persians, telling him that he would destroy Cyrus' power. Croesus' vexation and questions are highly admissible in view of his evaluation of himself as a pious man who constantly sends out treasures to the Delphi, which is also a gesture of boasting his wealth. The Pythian reply, in which the principle of inherited responsibility, although unexplained and at once self-justifiable, represents the preordained course of fate, closes the whole episode on the lot of Croesus and his Lydian monarch.

> No one may escape his lot, not even a god. Croesus has paid for the sin of his ancestor of the fifth generation before, who was led by the guile of a woman to kill his master, though he was one of the guards of the Heraclidae, and who took to himself the royal state of that master, to which he had no right. (*Histories*, 1.91.1)[21]

The revenge that has been prophesied to arrive inevitably (1.13.2: τίσις ἥξει) sees its eventual fulfillment with its disclosure for the purpose of enlightenment of the ever-imperious mind of Croesus. The undeserved reign of the Mermnad dynasty receives their doomed fall, conditioned by the fault committed five generations earlier. The oracular response further states the fact that Apollo's intention to delay the fall of Sardis to the lifetime of Croesus' son was not granted by fates (1.91.2: οὐκ οἷόν τε ἐγίνετο παραγαγεῖν μοίρας). But, as a return of Croesus' favor to the Delphi, Apollo postponed the taking of Sardis for three years and saved him from burning. The paradigmatic fall of Croesus and his rich empire, together with Solon's visit, offers Herodotus a narrative pattern on the subjects of the essential meaning of happiness, the uncertainty of human fortunes, the retribution of transgressing the *nomos*, and most crucially, the eradication of Eastern monarchies that is heavily dependent on hereditariness.[22] The notion that retribution will arrive sooner or later, to quote the metaphor of a spark growing into a raging fire in fragment 13.14–15, is indubitably an intertextual allusion to Solon.[23] Aside from this, the following observation shows more relevance to our concern in this section: the retributive effects of crimes committed by an ancestor and the personal fault of the victim, more specifically, his error of judgment caused by lack of insight, as explained successively in the oracle, matches well with the seemingly disparate components of fragment 13.[24]

As we are about to see, the gradual expansion of the notion of hell in the landscape of literature and philosophy in the 5th century BC marks the dusk of the tradition of inherited responsibility at this age.

## 4. Unresolved Dilemma: The Rise of the Idea of Punishment in Hell

With the rise of the awareness of individual responsibilities, the theodicy of inherited responsibility faces constant challenges, like the one that we have seen in Theognis. The basic idea is simple. A person who remains faithful and honest should not be burdened with ancestral faults and infliction. As Dodds put it, "with the growing emancipation of the individual from the old family solidarity, his increasing rights as a judicial "person," the notion of a vicarious payment for another's fault began to be unacceptable. When once human law had recognized that a man is responsible for his own acts only, divine law must sooner or later do likewise" (Dodds 1951, p. 150). At this point, the religious movements contribute to catering to the religious need for a way to make the evildoer suffer for his own penalty, even beyond the natural limit of a lifetime. Among them, the most notable one is the Orphic eschatology, which "assigns a key role to memory and even inverts conventional understanding: Hades is no longer a desolate region of oblivion; instead Earth is seen as the place of trials and punishment" (Mirto 2012, p. 29).

Scholars generally agree that the first Greek text that postulates the existence of a trial in the Underworld where people are punished or rewarded for their activities in their lifetime is found in Pindar. As we have mentioned in the first section, even though the idea of the Underworld (*Nekyia*) is already present in the Homeric epics (*Odyssey*, XI), it is generally depicted as the abode of the dead, which is grim and desolate, and no trace of a systematic post-mortem reward or punishment for the dead could be detected. Pindar's depiction shows a distinctive vision of the afterlife from the Homeric one.

> Upon sins committed here in Zeus' realm, a judge beneath the earth pronounces sentences with hateful necessity; but forever having sunshine in equal nights and in equal days, good men receive a life of less toil, for they do not vex the earth…in company with the honoured gods, those who joyfully kept their oath spend a tearless existence, whereas the others endure pain too terrible to hold (τοὶ δ᾽ ἀπροσόρατον ὀκχέοντι πόνον). (*Olympian*, 2.58–77)[25]

Pindar's clear articulation about the punishment and rewards the dead receive in hell should not be considered a sheer innovation *ex nihilo*.[26] The mystery cults that could be detected in the literature from the 7th century BC to the 5th century BC play a significant part in the final formation of punishing hell, most significantly with the concept of reincarnation in the Underworld. Starting from the 5th century, Pindar, Aristophanes, and Plato depict the afterlife in which there is a clear distinction between the good, who, after death, are ushered into a glorious afterlife, and the wicked.[27]

In the Eleusinian Mysteries, initiations are held every year for the cult of Demeter and Persephone based at the Panhellenic Sanctuary of Eleusis in ancient Greece; the eschatological principle is already present.

> Blessed is the mortal on earth who has seen these rites,
>
> but the uninitiate who has no share in them never
>
> has the same lot once dead in the dreary darkness. (*Homeric Hymn to Demeter*, 480–82)[28]

Archaeological finds also attest to the belief that a better afterlife might be won by initiation into mystery doctrines. The gold tablets in the shape of ivy leaves found in tombs in Thessaly, Macedonia, Crete, and Southern Italy (notably in Thurii, Petelia, and Hipponion) convey instructions for the deceased, directions for the journey to the underworld and words of affirmation said to the gods as proof of purification. Based on the different purposes of the tablets, Giovanni Pugliese Caratteli categorizes them into two groups (Carratelli 2003, p. 232). One group focuses on the instruction given to the dead; the other promises salvation to the dead. A tablet from a female tomb at Hipponion dated near the end of the 5th century BC is a characteristic example of the former group.

> This is the work of Memory, when you are about to die down to the well-built house of Hades. There is a spring at the right side and standing by it a white cypress. Descending to it, the souls of the dead refresh themselves. Do not even go near this spring! Ahead you will find from the Lake of Memory, cold water pouring forth; there are guards before it. They will ask you, with astute wisdom, what you are seeking in the darkness of murky Hades. Say, "I am son of Earth and starry Sky, I am parched with thirst and am dying; but quickly grant me cold water from the Lake of Memory to drink." And they will announce you to the Chthnoian King, and they will grant you to drink from the Lake of Memory. And you, too, having drunk, will go along the sacred road on which other glorious initiates and bakchoi travel.[29]

In several of the so-called "Orphic" gold tablets, the deceased is told that she will enjoy celebrating rituals in the Underworld, having earned that privilege by winning favor with the gods and being buried with the token of that privilege, the thin piece of gold foil with the text, a typical instance for the latter group is a gold tablet found in a tomb at Pelinna, in Thessaly. In it, the dead soul implores Persephone to welcome her to the blessed abode

in recognition of her innocence after she has served her punishment for her misdeeds. In recompense for the actions she has done, she asks for an eternal blessed life.

> Pure I come from the pure, Queen of those below the earth, and Eukles and Eubouleus and the other gods and daimons; For I also claim that I am of your blessed race. I have paid the penalty on account of deeds not just; Either Fate mastered me or the lightning bolt thrown by the thunderer. Now I come, a suppliant, to holy Phersephoneia, that she, gracious, may send me to the seats of the blessed. (Graf et al. 2007, p. 15)

In contrast with Pindar's idea of a trial of souls, both groups of tablets indicate that the initiation into the mysteries only aims to offer some hope of a more privileged destiny in the Underworld than the uninitiated. Nonetheless, the mysteries do not aim to offer any dogmatic teachings. The cult officials preach no particular views concerning the post-mortem fate of the initiated, but rather, the belief in a blessed afterlife that came to be associated with the mysteries grew from various initiates' attempts to understand their own experience of the rites (Bowden 2010, pp. 47–58). It focuses on the salvation of the purified soul since, through the sacred rites, the soul acquires freedom and returns to the blessedness of its original divine condition. Such salvation is certainly made in contrast with the Homeric world, where "the soul released from the body was credited only with a poor, shadowy, half-conscious existence so that an eternity of godlike being in the full enjoyment of life and its powers was only thinkable if the body and the soul, the two-fold self of man, were translated in undissolved communion out of the world of mortality" (Rohde 1925, p. 345). The gist of salvation lies in the purification of all sins and the subsequent relief from the cycle of birth and death, with the achievement of a permanent state of happiness in the other world. Plato's words are significant testimony to such a notion, although he himself discards ritual as the determining element for the fate of the soul in the construction of his version of hell.

> And does not purification consist in this, which was mentioned in the ancient word, to use all our means to keep apart the soul from the body, and teaching the soul the habit of collecting and holding on itself, away from all bodily elements, and remain, as far as it can, both in the present as well as in the future life, alone in itself, intent to its freeing from the body as from fetters? (*Phaedo*, 67d)[30]

The concept of purification in Orphism makes it possible for humans to expiate worldly sins. But what exactly are the sins requiring expiation? The content of "deeds not just" in the above passage invites debates. Some scholars tend to believe that the "Orphic" doctrine of purification involves "an original sin that would weigh on the human race" (Mirto 2012, p. 48). This biblical reading is strongly rebuked by recent scholarship, which argues convincingly that the fault requiring expiation involves "general, unspecified injustices committed by one's ancestors" (Mirto 2012, p. 48). This reading could be supported by Plato, who relies heavily on Orphic traditions. In the *Republic*, Plato says that Orphic priests "can expiate and cure with pleasurable festivals any misdeed of a man or his ancestors."[31] In the *Phaedrus*, Plato makes an even more explicit reference.

> Again, where plagues and mightiest woes have bred in certain families, owing to some ancient blood-guiltiness, there madness has entered with holy prayers and rites, and by inspired utterances found a way of deliverance for those who are in need; and he who has part in this gift, and is truly possessed and duly out of his mind, is by the use of purifications and mysteries made whole and except from evil, future as well as present, and has a release from the calamity which was afflicting him. (*Phaedrus*, 244d–e)

Thus, reincarnation through purification offers a solution to the problem of evil and divine justice in the late archaic period more self-sufficient than the inheritance of guilt or punishment after death. But as Plato's statement in the *Phaedrus* shows, the notion of inherited responsibility has not completely disappeared from the religious construction of punishing hell. On the contrary, it blends into the framework of post-mortem punishment

in hell by granting people opportunities to shake off their ancestral sins via participation in the mysteries and sacrificial offerings. E. R. Dodds' observation is crucial at this point:

> As for post-mortem punishment, that explained well enough why the gods appeared to tolerate the worldly success of the wicked, and the new teaching in fact exploited it to the full, using the device of the underworld journey to make the horrors of Hell real and vivid to the imagination. But the post-mortem punishment did not explain why the gods tolerated so much human suffering, and in particular the unmerited suffering of the innocent. Reincarnation did. (Dodds 1951, pp. 150–51)

To continue our discussion on Pindar, the Orphic concept of the dualistic nature of man certainly influences Pindar. The two elements of post-mortem judgment and a cycle of incarnations are later subsumed by Plato into his philosophical efforts, which greatly advances the final formation of the notion of hell in philosophy. Plato's understanding of hell is interwoven with the Orphic and Pythagorean doctrine of metempsychosis, and throughout his various writings, one principle remains firm: "That which gives meaning to this life is the soul's eschatological destiny, that is, the other life; everything here only has meaning if it is related to an afterlife, where the just and virtuous man is rewarded and the unjust and evil man is punished" (Edmonds 2013, p. 72).[32] Using the eschatological imagery, the Platonic Socrates alters the imaginative vision of the Underworld, as suggested by Homer, into a meaningful contemplation of philosophy. The most prominent example comes from three major texts: the myth of Er in the *Republic*, the affirmation of hell in the *Gorgias*, and the thorough description of hell in the *Phaedo*. All three eschatological myths appear at the end of these dialogues, replete with controversial claims about the right way to live. Nonetheless, it must be noted that the Platonic representations of hell in these works are by no means consistent and uniform (Edmonds 2013, p. 285). Different features are stressed in different works in accordance with their own needs. As Anthony Hoppers observes, Socrates' various descriptions of the Underworld are unified "neither through the details they offer concerning the architecture of the House of Hades nor through their accounts of the particularities of post-mortem existence of its residents." However, they are "fundamentally linked through more general compositional features, including their use of mythic discourse, their relation to the broader philosophical structure of the dialogue" and "their appropriation and reimagination of traditional imagery, most prominently concerning the visions offered of the Underworld itself" (Hooper 2021, p. 166).

Before discussing the Platonic hell, the proof of the status of life after death in the *Apology*, the speech on legal self-defense, which Socrates spoke at his trial for impiety and corruption in 399 BC, is a good point of departure for it. In this dialogue, Socrates tries to prove that death is one of two things: either annihilation or a continued existence in the Underworld. Thus, for either of them, we have no reasons to show fear: the annihilation is not to be feared, for there will be no consciousness of it; if it is the case of the latter, there is even less reason to fear it, since it is going to be complete blissfulness for the noble souls. Yet, such a proof is less valid as it seems to be. The journey to the Underworld is described in different ways, such as a change (40c7), a migration (40c), or a long journey abroad (41a5). The four judges seen scattered in the Greek myth are brought together in his version, namely Minos who is the judge of the dead in Homer (*Odyssey*, 11.568), Rhadamanthys who serves as an infernal judge in Pindar (*Olympian*, 2.55), Aeacus (in Aristophanes' *Frogs*, 464–77), and Triptolemus (in the Eleusinian Mysteries). The convocation of the infernal judges from different and sometimes incompatible traditions that are later attributed to Orpheus, Musaeus, Hesiod, and Homer (41a6–7) is meaningful. The inconsistent traditions of which Socrates is explicitly aware may incur skepticism since hell, if there is one, can only be the same. Furthermore, no clear sign of afterlife judgment could be drawn from this part. What is life in the Underworld like then?

> I am willing to die many times over, if these things are true; for I personally should find the life there wonderful, when I met Palamedes or Ajax, the son of

Telamon, or any other men of old who lost their lives through an unjust judgement, and compared my experience with theirs. I think that would not be unpleasant. And the greatest pleasure would be to pass my time in examining and investigating the people there, as I do those here, to find out who among them is wise and who thinks he is when he is not. What price would any of you pay, judges, to examine him who led the great army against Troy, or Odysseus, or Sisyphus, or countless others, both men and women, whom I might mention? To converse and associate with them and examine them would be immeasurable happiness. (*Apology*, 41a–c)

The description itself is strikingly nonhierarchical: the murderous Sisyphus has the same fate as the hero Ajax; Palamedes, who is drowned by Odysseus, has the same fate as the latter as a criminal. The egalitarian treatment of these distinct characters is far from being a valid proof of justice. Thus, these elements "at once unexpected and counterproductive" lead scholars to believe that the Socratic hell, where Socrates imagines himself to pass the pleasant time in conversation and to test their wisdom by close question and answer, is not a "single conception at all, but a jumble of different, incompatible ones," and that the hell itself should be viewed as an irony, instead of being a serious effort to establish a just world beyond (Benitez 2021, pp. 131–32). Whatever the case is, this early jumbled representation of hell awaits further innovative explorations, judging from the early date of the dialogue.

Among the Platonic innovations, the myth of Er is a prominent one. It is another representation of the Homeric *Nekyia*, a journey undertaken by a common soldier named Er to the Underground, only to be returned to life ten days later. Its resemblance to Odysseus' journey to the Underground is remarkable. Anthony Hooper offers a detailed comparison of Plato's version with the Homeric narration (Hooper 2021, p. 168). Several aspects are remarkably similar, which leads us to conclude that the Platonic description of hell is based on the Homeric framework. In the first place, the varying fates of people of prominence in the *Iliad*, including Telamonian Ajax, Agamemnon, Thersites, and even Odysseus himself (620a–d), receive considerable attention in both Homer and Plato. Furthermore, Er narrates how the departed sail in a grand cosmic trireme (615c), beyond numerous big whorls (615c, alike to those manufactured by Charybdis), and even confront Sirens in the course (617b). Moreover, distinct geographical elements particularly signify this account as Homeric in inspiration. Plato likewise mentions meadows (616b) and the Homeric Plains of Forgetfulness (621a3), accompanied by the water features of the Underworld (621a). Finally, Er, similarly to Odysseus, returns from the Underworld to tell of what he has seen.

The basic change happens with the introduction of an afterlife trial in hell, which is made explicit in the commencement of Er's narration. As we have said, the post-mortem judgment and the reality of reincarnation are essentially Orphic in origin. He begins with the judgment of the departed, in which the souls will be rewarded with an eternity in the heavens or a punishment in Tartarus, depending on their righteousness in life. The punishment the wrongdoers receive in hell is not actually commensurate with their wrongs. All the wrongs they had ever inflicted, and all they had wronged, had been repaid tenfold; a hundred years for each wrong, so that if human life was considered to last a hundred years, the punishment would be ten times the crime. Likewise, holy men will receive their due reward in the same measure. The sins of impiety towards gods and parents and of tyranny are considered to be the most severe ones. Ardiaeus the Great, a one-time tyrant in a certain city of Pamphylia just a thousand years before who killed his father and elder brother with many other unholy deeds, is singled out as a typical example of those who must undergo a thousand years of punishment in the hell for his sins to be completely expiated. The scene is horrifying. The guardians " bound his hand and foot and head and flung down and flayed them and dragged them by the wayside, carding them on thorns and signifying to those who from time to time passed by for what cause they were borne away, and that they were to be hurled into Tartarus" (*Republic*, 616a).

After seven days of lingering, the purified souls are obliged to proceed on their journey, and their next destination is the seat of the daughter of Necessity Lachesis. The mortal souls will behold a new cycle of life and mortality. Unlike the Orphic tradition that emphasizes the function of sacrificial offerings in deciding one's future destiny, in the Platonic version of reincarnation, the choice is entirely one's own, as the prophet of Lachesis has declared that "your genius will not be allotted to you, but you will choose your genius; and let him who draws the first lot have the first choice, and the life which he chooses shall be his destiny. Virtue is free, and as a man honour or dishonour her he will have more or less of her; the responsibility is with the chooser—God is justified" (*Republic*, 617e). With this statement, Plato highlights a significant point that the virtues and sins in one's lifetime will not perish as the body decays but will continue as crucial guides as one chooses his own way of the next life. The Platonic hell, drawing on the sources of the Homeric depiction of Hades, the Orphic idea of afterlife trial, and the Pythagorean notion of a geometrically ordered cosmos, is transformed from the mythical and religious imaginations into a serious play of philosophical reflection. The fact that Plato concludes his construction of an ideal state attests to the importance of the theocidy that is also the central concern of Solon's inherited responsibility—the evil will be punished, and the good will be rewarded, although the two authors make use of different approaches and traditions.

Such a construction of hell is also a strategy against the relativistic views held by contemporary sophists. Viewed from this perspective, the Platonic hell corresponds with Plato's efforts of the interiorization of justice, say in the *Republic* book IV, when faced with the challenge from the relativistic views of the sophists as Callicles and Thrasymachus, who deny the presence of gods ruling over the human world and go as far as to claim that injustice is more beneficial. In the *Gorgias*, Callicles, taking up with Polus the subject of whether doing injustice or suffering is more evil, goes on to claim that the laws of the polis must agree with human nature, which is characterized by its desire to get a greater share since nature itself makes it legitimate that stronger human beings get a greater share than weaker ones (483c1–e4).

The Platonic Socrates firmly believes that an undisciplined man is unhappy and should be restrained and subjected to justice. As the ultimate critique of Callicles' blatant hedonism, Socrates asks his interlocutors to give ear "to a right fine story, which you will regard as a fable, but I as an actual account; for what I am about to tell you I mean to offer as the truth". As we naturally expect, the story is about Cronos judging men just after they die. The good and righteous men will be sent to the Isles of the Blessed, and the godless and unrighteous men will be exiled to the prison of vengeance called Tartarus. While the mortals were alive with their clothes on, the case was judged badly because the judges were always fooled by their appearances. In light of this, Zeus made his sons Minos and Rhadamanthus from Europa and Aeacus from Aegina naked judges in the Underworld and stripped naked of the bodies of the dead. Socrates declares that he believes in it and deduces from it that death is the separation of body and soul. It is further affirmed by Socrates that each individual retains the qualities they had in life after death, and when the judge takes some potentate, they will find that his soul is marked with the marks of his perjuries and crimes since these will be stamped on his soul. Socrates' argument based on "truth" perfectly dissolves the nihilistic approach of the sophist by resorting merely to empirical observations of human life. By the faith in hell, the *Gorgias* conveys a sort of certain optimism: we should not be discouraged by the fact that we can clearly see the virtuous suffering and the wicked prospering around us since there will eventually be a judgment where everyone will receive what they deserve.

A more elaborate treatment of the Platonic hell, together with related notions such as the consistency and immortality of souls and the suffering of evil souls, is found in Plato's *Phaedo*. In this dialogue, Plato contends that knowledge is a recollection of the already known and proves that multiple incarnations of the soul, which, like the ideal form but unlike the body, is eternal and will with good behavior at last pass "to the place of the true Hades, which, like her (the soul) is indivisible, and pure, and noble, and on her way

to the good and wise god, whither, if god will, my soul is also soon to go" (*Phaedo*, 72c). The point of departure here, unsurprisingly, is the very same as the one in the *Republic* and *Gorgias*, and certainly with Solon: "for if death were an escape from everything, it would be a boon to the wicked, for when they die they would be freed from the body and from their wickedness together with their souls" (*Phaedo*, 107c).

In order to convince the audience of the existence of hell, Plato offers a remarkably long description of the geography of the Underworld. He first sets up the Underworld as a place whose flowing elements can influence where we live and establishes the Underworld as an area with influential liquid components that affect our habitation. Then Plato states that the hollowed underground areas where we dwell are interconnected and that water streams in both directions, like in mixing bowls (111d). He continues to describe three underground rivers that represent the four elements: rivers of fire, rivers of water (hot and cold), and rivers of muck, both cleaner and dirtier (111d–e). Finally, he relates the geographical details of the Underworld with the afterlife punishment of the dead.

> Now when the dead have come to the place where each is led by his genius, first they are judged and sentenced, as they have lived well and piously, or not. And those who are found to have lived neither well nor ill, go to the Acheron and, embarking upon vessels provided for them, arrive in them at the lake; there they dwell and are purified, and if they have done any wrong they are absolved by paying the penalty for their wrong doings, and for their good deeds they receive rewards, each according to his merits. But those who appear to be incurable, on account of the greatness of their wrongdoings, because they have committed many great deeds of sacrilege, or wicked and abominable murders, or any other such crimes, are cast by their fitting destiny into Tartarus, whence they never emerge. Those, however, who are curable, but are found to have committed great sins—who have, for example, in a moment of passion done some act of violence against father or mother and have lived in repentance the rest of their lives, or who have slain some other person under similar conditions—these must needs be thrown into Tartarus, and when they have been there a year the wave casts them out, the homicides by way of Cocytus, those who have outraged their parents by way of Pyriphlegethon. (*Phaedo*, 113d–114b)

In the hell depicted in the *Phaedo*, the classification of the dead souls on a moral ground is much more specified. Unlike in the *Republic* and the *Gorgias*, the judgment in the *Phaedo* actually happens before the souls of the deceased depart for the Underworld. Once they arrive at hell, the curable ones will have to pay the penalty as is deemed appropriate for the sins committed when alive, while the deadly sinful ones will be cast into Tartarus, suffering eternal damnation. This is not a process of purification as one may expect it to be since their mortal qualities will not show any change through their deserved punishment. When they have received their due and remained through the time appointed, another guide brings them back after many long periods of time to the Acherusian lake (107e), where they will be sent back to be born again into living beings (113a), which seems to be the only indication of reincarnation in the *Phaedo*.

Nonetheless, the notion of reincarnation (not seen in the *Gorgias*) and the possibility of escaping from it (not seen in the *Republic*) elaborated in the *Phaedo* bring basic difficulty for it to be reconciled with the notion of afterlife punishment and for the constructive plan of hell. In this dialogue, Plato goes to great lengths to demonstrate the superiority of the soul over the body. For the soul, the body resembles a prison in which various desires manipulate reason and prevent it from exercising its force independently (82e–83c). The attachment of the soul to the body is the very sin requiring expiation. He explains that the good soul departs without difficulty from the body at death, while the bad soul clings to it and is driven by its need for manifestation to inhabit other bodies that match its former life (80d5–82c8). Since embodied existence itself is considered a flaw in contrast with the free existence of pure souls, reincarnation is perceived as a punishment for a bad life, and the utmost level of virtue is said to be held by the philosopher, who, by abstaining from

bodily pleasures, renders his soul pure upon death, thus freeing it from the cycle of reincarnation, going "into the invisible, divine, immortal, and wise, and when it arrives there it is happy, freed from error and folly and fear and fierce loves and all the other human ills, and as the initiated say, lives in truth through all after time with the gods" (81a). In other words, the mediocre souls that are strongly attached to the body will have to receive two rounds of punishment and the incurable will be condemned with eternal punishment in the Underground, while the disembodied enjoy permanent happiness somewhere unknown, and the less perfect ones will dwell on the true earth, "the earth itself, pure and set in the pure heaven in which the celestial bodies are" (109b7–8).

> Those who are found to have excelled in holy living are freed from these regions within the earth and are released as from prisons; they mount upward into their pure abode and dwell upon the earth. And of these, all who have duly purified themselves by philosophy live henceforth altogether without bodies, and pass to still more beautiful abodes which it is not easy to describe. (*Phaedo*, 114c)

Consequently, the views on the soul and the body and the system of reward and punishment based on such a view in the *Phaedo* change the whole cosmological vision, including the location of hell, following the kinship argument that the "soul's composition determines both what environment it is drawn to and how it interacts with this environment" (Ebrey 2023, p. 200). In the *Phaedo* 109a–110a, Plato offers a detailed visionary description of the cosmos from the perspective of a philosopher. The basic claim is that we humans dwell in a hollow in the earth, not on the surface of the earth itself, unable to see the surface of the true earth, as the fish is not able to see the surface of the sea.

> In this land, because it is of such a kind, the growing things, trees and flowers and the fruit, grow in proportion. And then again the mountains likewise have stones that are also in the same proportion with respect to their smoothness and clarity and beauty of their colour. And the little stones here that are most admired, the sards and jaspers and emeralds and all such gems, they are fragments from these. But there is nothing there that is not such and still more beauteous than these. (*Phaedo*, 110e–111a)

In this description, the earth itself is essentially beautiful, and it is "the real heaven, the real light, and the real earth" (109e). The whole regions where the mortals live ("this earth of ours," 110a), however, are injured and corroded, as in the sea things are injured by the brine, and nothing of any account grows in the sea, and there is nothing perfect there, but caverns and sand and endless mud and mire. The fluctuation of the four elements within the earth generates contamination in our dwellings, resulting in ugliness and sickness (112c3). Given the Socratic logic in the *Phaedo* that immateriality transcends corporeality, hell should certainly be much more undesirable than the dwelling place of humans. After explaining the force that brings rivers up from Tartarus, Socrates turns his attention to the four majestic rivers that serve as the structure of the Underworld with two sets of opposing pairs, Ocean being the first. According to Homer, the Ocean is an ever-running, encircling waterway of the earth (*Odyssey*, 11.13). Hades is exterior to Ocean (*Odyssey*, 10.508–12, 11.13–22). But Socrates reverses this, proposing Ocean as an everlasting, circular river, which incorporates Tartarus and the realm of the dead. The Acheron travels in the opposite direction to Ocean, leading to the Acherusian lake. It is here that those who lead ordinary lives come to rest after death (112e–113a). These two rivers divide the realm of the dead and chart the path of many after death. The other two rivers, Pyriphlegethon and Cocytus, will be the place where those who have done serious but reparable wrongs are punished.

> All the rivers flow together into this chasm and flow out of it again. And each river comes to be like the kind of earth through which it flows. The source of all the streams flowing out from here, and flowing inside, is that this liquid has no base and no foundation. So it oscillates and surges up and down, and the air and the wind around the liquid do the same. (*Phaedo*, 112a5–b4)

One of the most striking characteristics of such hell is the lack of stability caused by the ever-running rivers. It has been shrewdly observed that Plato is making use of the Heraclitean flux here: "flow" and "up and down" (Ebrey 2023, p. 287). The rivers constantly flow in and out, surging up and down, and the air and the wind do the same thing. All the descriptions point to the basic attribute of the punishing hell: nothing there is sound and true. The Platonic cosmology thus gives a clear geographical representation of "the earth itself," "the earth of ours," and the Underworld. The whole picture is as follows: the surface of the earth itself is pure and relatively free from the churning of elements, with the things there more beautiful than "the earth of ours"; we live between the "earth itself" and the Underworld, that is the interface with some beauty pouring down from above but at the same time receding due to the flow coming up from below; the Underworld, due to its unceasing activities of mixture and flowing, never shows a moment of stability.

Then, the problem arises with regard to the final destination of the disembodied philosophical souls, among which Plato assuredly considers Socrates to be the perfect model. Earlier, Plato mentions that the soul that has passed through life in purity and righteousness finds gods for companions and guides and goes to dwell in its proper dwelling (108c), i.e., "the earth itself." Then, what fairer place should the philosopher, with complete detachment from the body, have as his final residence? This is the very dilemma Plato is faced with regarding the further differentiation of the good men and the problem of corresponding punishment in hell (Annas 1982, pp. 127–29). Suppose the philosophers' reward is the final disembodiment. In that case, the second-best reward of the non-philosophical good will have to represent some kind of embodiment, which is represented as both a repulsive fault and punishment in the *Phaedo*. Consequently, the notion of reincarnation as a punishment conflicts with the simple tradition of afterlife trials in hell. If the completely pious men, yet without coming up to the Platonic standard of being entirely detached from the material world, will receive their punishments in hell for their clinging to the body, the faith in justice may suffer agitation. But if Plato follows the tradition of granting both of them the same place in hell, such as the Elysium, his whole emphasis on the superiority of soul over body will eventually fail.[33] As an inevitable result of this, Plato has to dodge the problem by saying that the place where the souls philosophically purified will finally go is "not easy to describe" (114c).

Drawing upon sources such as the Homeric traditions and the Orphic beliefs, three myths of hell are meaningful in different ways: in the *Republic*, the "systematic and dramatic reimagining of the Homeric foundations of the House of Hades and its extensions" (Hooper 2021, p. 169) stresses the values of cultivating the morality in life in forming good citizens, rather than those of gaining privileges in the afterlife by conducting certain rituals; in the *Gorgias*, the reassuring afterlife judgement is used to make the immoral Callicles desist on his proposition of the law of the strongest; in the *Phaedo*, the hell that is geographically revised and theoretically rewritten serves to justify the lack of fear of Socrates facing imminent execution and to console his disciples. Regardless of their marked differences and inconsistencies, the Platonic hell, as one of the two major ways to envisage justice and punishment, together with the notion of the afterlife and reincarnation it promotes, represents the final stage in the constructive history of hell in antiquity.

## 5. Conclusions

As we have seen, the Greek theodicy is pulled in two different ways, which, in general, try to go beyond the limit of a lifetime and find a possible approach that makes the sinners compensate for crimes that go away unpunished. The notion of inherited responsibility, explicitly articulated in Solon, Herodotus, and the tragedians, prefers to think in a way that justice is done by the transmission of the punishment to the guiltless descendants of the transgressors: either the man's children or his descendants thereafter. Solon's statement is the first explicit one in Greek literature. With the rise of self-awareness and the emancipation of individuals from family solidarity, people started to look elsewhere for a more satisfactory approach that harms no innocent ones, and such a notion was gradually

superseded by the idea of hell, whose tradition could be traced back as early as Homer. However, as we have shown, the notion of hell is not monolithic. For the convenience of understanding, the evolution of the Greek hell could be tentatively categorized into three phases: the Homeric, the Orphic, and the Platonic. In the Homeric phase, hell is simply a place where shadowy souls convene, with a few divine exceptions. The Orphic phase stresses the possibility for the initiated dead to have a more privileged life in hell than the uninitiated. The Platonic phases try to incorporate all those elements into a single system that aims to lead citizens into a moral life. The Solonian idea of inherited responsibility prevalent in the 6th and 5th centuries is analyzed in detail concerning the conditions for its emergence, the influence of the natural philosophy, and its basic difficulty, which is revealed by Theognis' complaining verses and causes the eventual dominance of the notion of hell. The third part tries to show how the Homeric Hades, a vision for the Underworld that shows no sign of afterlife trial, is used and innovated by later authors such as Pindar and Plato, who borrow the religious elements from the Orphic beliefs of eschatology and concept of reincarnation, with the intention to establish a self-contained system of justice and punishment. The evolution of hell culminates in the ones constructed by Plato in the dialogues of the *Republic*, the *Gorgias*, and the *Phaedo* to justify the benefits of living a moral life and the horrifying consequences of living an immoral one. Interestingly, within the Platonic framework, the vision of hell undergoes the same three phases without clinging to the details: the hell in the *Apology* resembles that of the Homeric one; the one in the *Republic* and *Gorgias* leans more towards reincarnation and afterlife judgment; the *Phaedo* represents the proper Platonic version of hell.

**Funding:** This research received no external funding.

**Institutional Review Board Statement:** Not applicable.

**Informed Consent Statement:** Not applicable.

**Data Availability Statement:** Data are contained within the article.

**Conflicts of Interest:** The author declares no conflict of interest.

## Notes

1. The Christian hell is certainly an important reference in this context. However, in the context of the Greeks, the notion of hell is historically unstable, which, as a matter of fact, applies to Christian hell as well. It could be a place where dead souls have their afterlife existence, a location of transfer for the souls to transmigrate, or a horrifying place where the souls of sinners are tortured and thus purged. In this paper, the word hell is used in multiple distinctive contexts, but its true meaning depends on its religious or philosophical context.

2. Cf. (Gagné 2013, p. 205). Through the evidence in the lyric poetry, he concludes that inherited responsibility "plays a much more prominent and varied role" in the 6th century compared to the early archaic period.

3. Based on this assumption, Anthony Hopper deduces the reasons for the lack of details with regard to the landscape in Hades since the Homeric underworld is simply a dark place, in contrast with the sunlit "upper world."

4. All translations of Homer are adapted from Lattimore (1951, 2007).

5. It needs to be noted that the early representation of the Isles of the Blessed is not as virtue-oriented as in later traditions. In the Homeric epics, the Elysium is a place reserved for three groups of people: the early heroes like Kadmos, Lykos, and Rhadamanthys; the Trojan war heroes such as Menelaus; the Elysian-born like Euphorion, son of Achilles.

6. Later reflections on inherited responsibility are made by Proclus' *De decem dubtationibus circa Providentiam* and Plutarch's *De sera numinis vindicta*. The latter has a wider audience. This essay is divided into three parts: the first part is concerned with the delay of vengeance within the span of one's lifetime; the second discusses the transmission of punishment to the descendants of the transgressor; the third explores the metempsychosis and shows that real punishment happens after one's death. This valuable document could be considered as evidence of the co-existence of the notions of inherited responsibility and a punishing hell in the religious thoughts of antiquity.

7. The first extant Greek text that postulates the existence of a trial in the underworld wherein people are punished or rewarded for their actions in the world is that of Pindar's *Olympian*, 2.58–77. This passage is incorporated into our later discussion of the Platonic hell. Judging from the texts of Solon that we have now, there is no clear representation of hell.

8    All Greek citations of Homer are from *Homeri Opera in Five Volumes* (Homer 1920), edited by David. B. Munro and Thomas. W. Allen.

9    Aside from the curse of collective punishment, the allegory of the prayers or *Litai* in the *Iliad* Book 9 should also be considered as a prelude to the articulation of Solon: although they are lame and sometimes late, and they exact punishment on living offenders, it is not excluded that they might do so to offspring.

10    All translations and Greek quotations of Solon are based on Douglas E. Gerber (1999)'s edition.

11    The resemblance of Ionian philosophy with Solon's thoughts has been accepted and developed by many scholars since its ground-breaking articulation by Werner Jaeger. Cf. (Jaeger 1966, pp. 89–93; Vlastos 1946, 1947; Gentili 1975). The resemblance suggests a shared philosophical tradition between the Ionian natural philosophers and Solon that becomes best known through the work of Anaximander. This speculation is further advocated by Maria Noussia and Joseph Almeida. Aside from their similar understanding of justice as a cosmological recompense, Noussia points out that Solon's traveling, which could be well attested in both early and late sources such as Herodotus and Plutarch, reinforces the possibility of his knowledge of the Ionian philosophy. Cf. (Noussia 2006, p. 144; Almeida 2003, pp. 71–85).

12    English translation by Glenn Most (2018). Also cf. Pindar, *Nemean Odes*, 4.32: it befits one doing something also to suffer; Aeschylus, *Libation Bearers*, 309-15: "Let evil tongue be paid for evil tongue. In doing what is due Justice cries loudly, 'Let one pay bloody stroke for bloody stroke.' 'That the doer suffer' is a story thrice-told".

13    Adapted from J. M. Edmonds' translation.

14    In archaic Greek, the word ἀναίτιοι is related to αἰτία, whose primary meaning is "cause." Therefore, this word, if taken literally, means "without causal responsibility."

15    The fragment is preserved by Simplicius (*Physics*, 2.4.13). The translation is Werner Jaeger's, cf. (Jaeger 1947, p. 34).

16    Jaeger rejects what Diels proposes for the interpretation of τάξις, i.e., "order." Drawing upon Greek phrases such as τάττει δίκην, τάττει ζημίαν, and τάττει θάνατον, Jaeger thinks that τάξις must mean "ordinance," rather than the more general meaning of "order" (Jaeger 1947, p. 207).

17    For a comparison of the views expressed by the Herodotean Solon with those of the historical figure, cf. (Chiasson 1986, pp. 249–62). The agreement between the historical figure Solon and the Herodotean Solon has been endorsed by scholars in general. For instance, Waters argues that opinions cannot be assumed to be Herodotus' own unless he expresses them in his own persona. Thus, Waters concludes that the Herodotean characters should be seen as "those he thought suitable to the occasion," and "Herodotus makes Solon say what is appropriate to Solon" (Waters 1985, pp. 104, 99).

18    All Greek translations of Herodotus are from A. D. Godley (1960)'s edition.

19    cf. (Lateiner 1989, p. 142). According to his study, "seven violators of supra-national *nomoi* are said to be childless, at least in the male line: Astyages, Cambyses, Cleomenes, the elder Miltiades, son of Cypselus, Stesagoras, son of Cimon, and the legendary Polybus and Cepheus."

20    The two Delphic oracles are explained by the Phythian priestess in 1.91.4–6. For the destruction of a great empire, the oracle referred to the one of Croesus himself, who missed the mark due to his recklessness. For the mule as the king of the Medes, the oracle alludes to Cyrus, whose mother was "a Mede and the daughter of Astyages king of the Medes" and whose father "was a Persian and a subject of the Medes and although in all respects her inferior he married this lady of his." All this points to Croesus' fall at the hands of Cyrus.

21    Translation by A. D. Godley.

22    The principle of total eradication, as shown in the case of Croesus, is also displayed in other Eastern monarchies. The Greek word for total destruction πρόρριζος appears three times in Herodotus. Cf. (Lateiner 1989, p. 144): "the word πρόρριζος at 1.32.9 implies that the loss of happiness includes the loss of one's descendants". Its first appearance is in Solon-Croesus logos, as is shown in our discussion. The second appearance is in Herodotus, 3.40. 3, where Amasis sends Polycrates a letter on the envy of the gods for those who are consistently fortunate and the outcome of total destruction of a household: "For from all I have heard I know of no man whom continual good fortune did not bring in the end to evil, and utter destruction." The third and last appearance of the word πρόρριζος in Herodotus involves loss of descendants as well, and it points to the critique of the Spartans. Cf. 6.86δ where Leotychides asserts that Glaucus, who breaks the oath, would suffer retribution that his family line in Sparta will be completely exterminated "down the roots" (πρόρριζος).

23    On Herodotus' knowledge of Solon's verses, cf. Chiasson makes the confident declaration that "Herodotus consciously and explicitly evokes the memory of Solon's verse. Otherwise, although Solon's speeches contain no compelling verbal echoes of the poetry, the conceptual affinities between them are sufficiently striking to suggest that Herodotus knew Solon's poetry well and attempted, with remarkable historical consciousness, to incorporate its most prominent themes into the speeches he composed for the Athenian" (Chiasson 1986, pp. 250–62).

24    What makes the role of the notion of inherited responsibility in Herodotus even more interesting is the aforementioned coexistence with the myth of Hades. In Book II of the *Histories*, Herodotus recounts the story of the Egyptian pharaoh Rhampsinitos, who is reported by the priests that he goes down to the place called by the Greeks "Hades" and plays dice with Demeter. Then, he goes on to tell the audience about the Egyptian religious belief in an obvious digression of the transmigration of souls. Ac-

cording to his report, Demeter and Dionysos are rulers of the world below; the Egyptians are the first to teach the doctrine that the soul of humanity is eternal, and upon the decease of the body, the soul is reborn into another being that has just been born, and after cycling through all the creatures of the land, sea and air, it once again inhabits a human body as it is brought into the world; and that this cycle takes place every three thousand years. Herodotus omits the details concerning the punishment and reward during transmigration, but his critique of the Greek appropriation of the originally Egyptian doctrine gives us a significant clue to the source of the idea of punishing hell.

25   The English translation belongs to W. H. Race. On the possible Orphic origin of the theory of transmigration, cf. (Jaeger 1947, pp. 85–87; Edmonds 2013, pp. 248–95).

26   Inherited curses, as we see in Aeschylus' *Oresteia* and other tragedies concerning the Labdacids, are indeed an important facet of the notion of inherited responsibility. However, it seems to be less relevant to our present discussion, which focuses on the mechanism of justice and punishment. My forthcoming book on inherited responsibility has a chapter discussing this issue in detail.

27   The possibility of the Orphic influence on Pindar has been advocated by many scholars. M. M. Willock (1995), for instance, finds traces of Orphism in Pindar through their resemblances in language. Also cf. (Currie 2005, pp. 389–90; Catenacci 2015, pp. 15–32). He discusses the instances in Pindar talking to an audience with special knowledge. A note must be made on the so-called Orphism. The definition of it has always been controversial due to our vague knowledge of this religious phenomenon and its complex constituents of religious beliefs and mythical tales, elements of worship, practices, and lifestyles. Its relation with the cult of Dionysus, with the mysteries of Demeter, and with Pythagoreanism makes its definition even more difficult. In this paper, my discussion of Pindar is confined to the most characteristic and unique beliefs of Orphism: metempsychosis and belief in afterlife punishments and rewards. Edmonds' detailed study of Orphism is also a valuable source for this subject (Edmonds 2013, pp. 6–10).

28   H. Foley's translation.

29   (Graf et al. 2007, p. 5). Translation by S. I. Johnston.

30   All Greek quotes of Plato are based on John Burnet's edition (Plato 1903). All translations of Plato are based on *Plato in Twelve Volumes* (Plato 1966), with some changes if necessary.

31   *Republic*, 364b.

32   The Pythagoreans created a complex religious doctrine that entailed theories regarding music, mathematics, diet, and ascetic practices. They thought that souls were reborn and passed on and are often credited as the originators of Platonic doctrines. In late antiquity, the Orphic and Pythagorean beliefs merged together. This combination is typically referred to as Neo-Pythagoreanism.

33   Other than this, what makes the whole matter of reincarnation even more problematic is that there is nearly no sign that through the process of reincarnation, the quality of each life has an effect on the quality of the subsequent one.

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
