# Peer review of "“Mills of God”: Two Ways of Envisaging Justice and Punishment in Greek Antiquity"

_religions, doi:10.3390/rel14121549_

Round 1

Reviewer 1 Report

Comments and Suggestions for Authors

The article overall could use some rethinking as to the connections and the breadth of the material; the chronological span from Homer to Plato with two major concepts (underworld & whether or not a punishing hell; ancestral guilt & justice) requires more space than the 24pp. of the mss. to treat properly. The author(s) should seriously consider revising so to focus on the Homeric underworld and Solon's concept of ancestral guilt in response to it, cutting everything from section 4 on. This will help address the concerns below about how coherent the article is as a whole.

Additionally, the article is disjointed. There is included Greek text in about half of the article but not the other half, and there is untranslated Greek in part of the mss. but not all. There are no clear connections drawn between the sections on the underworld (Homeric or otherwise) and ideas around ancestral guilt. This includes the most obvious, that the concept of ancestral guilt developed to fill the place ideas of a punishing hell usually hold.

There is a propensity to apply Classical or Hellenistic interpretations of justice and responsibility to Solon's poetry; this needs to be revised so that Solon is discussed in his capacity as an archaic poet and thinker. Ovrall the author(s) needs to be more careful with context and dates; see esp. the comments below on Anaximander and Solon. 

Below are specfic comments/ queries noted by line number. 

l. 36: ancient Greek mind, surely, is what is meant here

l. 44: the author(s) should address what is meant by "hell" in reference to ancient Greece, particularly as the concept of the afterlife is categorically different than commonplace definitions of hell

ll. 135: book number missing from citation

FN 1 on p. 3: further elaboration is needed for the FN to support the discussion

l. 168: there is no citation to Sourviou-Inwood

l. 183ff: Isles of the Blessed as a reward for virtue: originally Menelaus gets to go to the Isles simply because he is the husband of Helen, who goes because she is the daughter of Zeus. When the Isles are discussed in earlier versions of Greek mythology and mythography, they recognize stature in life, not necessarily virtue; as the author(s) is discussing the Homeric underworld and not the late Classical or Hellenistic, when the idea of a moral right to the Isles of the Blessed becomes more pronounced, this early tradition around the Isles needs to be addressed.

l. 190: typo with Leto

l. 219: "the former interpretation" unclear

l. 223: Who expects Hades to be a judge of the dead? He is not connected with the act of judging, even as the below-the-earth pair to Zeus, unless the authors have archaic/ early classical evidence otherwise

l. 276: typo with Lloyd-Jones

l. 283: inherited responsibility = a partial to full denial of the existence of Hades: this argument relies on the assumption that an underworld or hell must require punishment, but as the author(s) has just demonstrated, this is not the case for the Homeric underworld. Solon no where in his verses denies the existence of Hades (and not mentioning Hades does not equal a denial due to general practice of not bringing attention to oneself by mentioning the King of the dead or the god of the dead); this argument needs to be reconsidered in line with Solon's corpus as a whole and Greek attitudes towards the underworld, discussing the underworld, and inherited guilt in the archaic period.

l. 314ff: it may be useful to bring in the Erinyes as the protector of oaths here

l. 329ff: does Hades ever become a space equivalent to a punishing hell in the Greek tradition, outside of the Platonic tradition which is conveying its own ideas about philosophy and the soul?

l. 379f.: The connection between Anaximander and tisis needs to be spelled out more. The dates need to be reconsidered as well: Solon's poetry is generally dated to his archonship (594 BC); Anaximander was born in 610, meaning be would have been about 16 when Solon's poetry was dated. As disjointed as the poetry is in Solon 13, the thought is mature; so how then, did a 16 year old Anaximander develop the concept of tisis fully enough to influence a fully mature Solon at the height of his political career?

l. 389 and passim: use Greek or don't for single words like tisis

l. 409: what is the central argument

l. 422: typo with 208)4

l. 453: in archaic Greek, aitia was the sense of "responsible for, causal" so that anaitioi = without causal responsibility. The distinction is slight from English "innocent" but it should be noted as the point is that those who did not cause the guilt are being punished for it.

ll. 546-47: Herodotus is active in the 5th century, not remains

l. 615: check Greek formatting

l. 631: typo, Solon for Croesus

l. 665: Herodotus' concern with monarchic rule is the slide into despotism, which can affect Greek tyrannies as much as hereditary dynasties; "the radication fo eastern monarchies" point needs to be better supported if it is kept

ll. 673on: There are significant problems with Herodotus' treatment of Egyptian customs and religion; they need to be addressed and the author(s) needs to engage with the substantial body of scholarship on the issue 

ll. 717on: explain how  the mystery cults contribute to a punishing hell and what evidence there is for this, since the whole point of a mystery cult is the practices and beliefs are not shared with outsiders. Further, what we do know of mystery cults suggests that the focus is on a special underworld for initiates that is better than the Homeric underworld, which does not necessarily mean a punishing hell.

l. 766: do the mysteries teach how to access the special part of the Underworld for the initiates?

Orphism discussion: the author(s) need to address the connection between Orphism and mainstream Grek religious thought re: an underworld and afterlife

l. 819: dates for Orphism need to be engaged with the support contention that Pindar influenced by Orphic thought

l. 880: how is "a just world" being defined?

ll. 931-34: the argument about Plato and Solon is unclear

ll. 1052-53: typo in linebreak

ll. 1073ff: line break issues, Phaedo not italicized

Comments on the Quality of English Language

Good

Reviewer 2 Report

Comments and Suggestions for Authors

The article is well written, it is very clear in its structure.

The development of this role is consistent with the initial hypothesis. This is well defined, as well as the objectives of the work. The conclusions summarize well the value of this contribution in a field of study between philosophy and religion.

Reading is easy because it is well written and agile.

  The article is very interesting to learn about the two-way punishment, also post-mortem, for transgressions committed, precisely assessing this motive in each author (Homer, Solon, Plato), literary genre (epic poetry, elegy, Platonic dialogue) and era (from aracism to the classical period).

The author clearly points out the two ways that occurred in the ancient Greek world to configure the idea of responsibility imposed on the individual and justice, divine, received in return, punishments and rewards.

The Greek texts cited have been well chosen, and the citations and references to secondary bibliography are appropriate. The authorship of the translations is not indicated and perhaps should be added.

Reviewer 3 Report

Comments and Suggestions for Authors

The paper examines ancient Greek responses to the moral problem that is posed by the fact that sinners get away with murder and much else.  Two solutions are presented: the idea of hell, where offenders are punished, and that of inherited guilt, where punishment falls on descendants.  The author argues for a three-stage development (this could be stated more clearly in the introduction.  Early accounts of hell indicate that what survives there is a bare shade, not really subject to punishment.  In this vacuum, the idea of inherited guilt was expressed most explicitly by Solon, and taken up also by Theognis and Herodotus. When, however, the morality of punishing innocent descendants was questioned, new ideas of hell emerged.  On the religious side, there were the various doctrines subsumed under the label "Orphism," and in philosophy, Plato proposed a regime of punishments and rewards in the afterlife.

While I believe that the paper is publishable as it stands, I would urge the author to reconsider the chronology of Orphism.  Is there really any reason to suppose that it emerged after Homer, not to mention Solon (and by the way, it is not entirely certain that Homer precedes Hesiod)?  There is good reason to think that Homer excluded contemporary religious beliefs of many kinds, and Orphism may well have been among them.  Nevertheless, it may have come to figure more prominently in literature in the Fifth Century, as evidence by the Derveni papyrus, which seems to go unmentioned in the text.  Richard Janko has ascribed it to Prodicus!

On delayed punishment, mention should be made of the allegory of the prayers or Litai, in Iliad Book 9: they are lame and arrive late, and while they are presumed to exact punishment on living offenders, it is not excluded that they might do so to offspring.

I think a brief comparison with the Hebrew Bible might also be in order, where both doctrines - that of punishment unto the third generation and the idea that the innocent ought never to be punished - are present.  We may recall that Ezekiel affirms that sin is not transmitted from one generation to the next; if a father sins, it is wrong that his offspring be punished (18:4; cf. Deuteronomy 24:16 but contrast Exodus 20:5).

There seems to be no discussion of inherited curses, as in the Oresteia and many other texts.  Oedipus pays for the sin of Laius, etc.  Some reference to this tradition would be appropriate

All in all, the paper is quite long, and I do not expect the author to take up my suggestions at any length.  But I believe that they would improve the paper if at least given some recognition.

Some mention might be made of Plutarch's essay on why punishment comes late; of course, this is a later work, but it shed light on earlier ideas.

Round 2

Reviewer 1 Report

Comments and Suggestions for Authors

If the author(s)'s aim is to offer an outline for the two solutions for unpunished crimes, following Dodds, that aim needs to be more clearly stated in the introduction, and there then needs to be a restructuring of the article to support that intention. As it is, the article presents a very loosely connected review of ideas of the afterlife (section 2, 4, and 5) with a discussion of inherited responsibility (section 3) in the middle. Further, if Dodds' work is the guiding structure, then his arguments need to be presented clearly as the guiding structure for the article; the author(s) then needs to engage with critiques and developments of Dodds' work beyond Lloyd-Jones and lay out how the author(s)'s work is contributing to the conversation. As is, there is very little that is clearly new as opposed to a review, so that the contribution to scholarship simply consists of putting these conversations in the same room together, rather than actually getting them to talk to one another.

My initial concerns about the disjointed nature of the article have not been addressed or improved upon and so remain; the sections still do not talk to one another easily, and there is a distinct difference in citation usage and Greek text between sections. It appears that there are multiple authors who each wrote a section or two and then put them together. If there is going to be Greek text in one section, there needs to be Greek throughout (or, since there's no substantive engagement with the Greek aside from individual words here and there, take it all out as this is not a Classics journal).

My concerns about historical context and dates have only minimally been addressed (and see the comments below on n. 5). It is not sufficient to include a footnote saying that is it not "true to fact" that Anaximander influenced Solon yet maintain in the main body of the text that he did in fact influence Solon. The two seem to be working in a shared tradition of Ionian thought-- that is sustainable by the historical context and tradition of interelite intellectual communication via the symposion and travels; the author(s)'s argument around Anaximander, Solon, and tisis as stands is not sustainable by historical context.

My concerns about Herodotus on Egypt have not been addressed nor even engaged with in the author(s)'s report. There are significant issues with the Herodotean narrative on Egypt, to what purpose the various logoi are being put, and where Herodotus' information is coming from (let alone whether it is valid). If the author(s) want to maintain the point about Herodotus on Egyptian myths and the evolution of ideas on a punishing hell, the scholarship on Herodotus' use of Egyptian logoi and the problems inherent in them as well as the use to which they are put needs to be engaged with. If the author(s) is not willing to do that, then the paragraph should just be removed.

Individual comments that have arisen in the reread:

l. 205: what is the "typical afterlife" Homer departs from? Is it more correct to say that later versions of the myths around Tityos, Sisyphus, and Tantalus expand the rationale for the punishments? Again, the author(s) need to address the historical context as much as possible 

l. 279ff: missing citation to Lloyd-Jones

p. 6, n. 5: I am unclear on how the FN relates to Solon. Again, dates are an issue: Plutarch lived approximately 700 years after Solon, Proclus well over that; how are they then evidence for the archaic period and Solon's departure from the Homeric Hades? 

l. 286ff: the two sentences that finish this paragraph contradict one another unless the relationship between Solon's idea of inherited responsibility is explained in the particulars of how it connects to that in Homer/Hesiod as well as how it departs. A similar need to elaborate on connections and departures appears below, for Gagne on ancestral fault (l. 307ff) and the principle of inherited responsibility in Solon (l. 323ff); how is Solon's sense of inherited responsibility a departure from the Homeric model rather than a continuation of developing thought around ancestral guilt as seen with the curses against oath breakers? The author(s) is committed to making Solon's thought a radical departure-- what are the reasons for that and why does the author(s) not consider stressing the evolution in thought as a continued tradition from Homer to Solon to Anaximander? 

Paragraph beginning l. 292: the elaborate review of the duel is unnecessary to the discussion. Ditto for the elaborate reviews of the Croesus logos from p. 12 onwards. The large quotes from Plato also seem to be contributing little to the discussion

l. 366: This may be the author(s)'s revision to the comment made on the original l. 283; either way, if Solon expresses a theological framework that excludes hell, afterlife, or transmigration, the evidence for this exclusion needs to be provided. Or the author(s) need to address the point that there is a general reticence in archaic authors (outside of myth) to invoking any deity involved with the underworld as well as the fact that we do not have the entirety of Solon's corpus so caution is in order in conclusions

l. 382-3: I am unclear as to why a single sentence is functioning as a paragraph; is this supposed to contribute to the discussion of Anaximander and Solon?

l. 469ff: see comments above on use of Greek text

l. 478: if there are various interpretations possible but yet a near unanimous agreement as to what is meant, some scholarship on the matter would be nice to see

Paragraph beginning l. 513: the point is unnecessarily complicated in this paragraph, while the question of justice being insufficient (l. 523) needs to be articulated more fully

Paragraph beginning l. 547: the connection between the classical discourse on greed and its political ramifications needs to be connected to inherited responsibility more clearly

l. 557: language choice with Oriental

Paragraph beginning l. 564: the connection to inherited responsibility takes a while to get to (2/3s of the way through) and is weak, sustained simply by the use of "uprooting"

l. 666 and n. 20: the note added to my comment on the original l. 665 does not support the argument about "the eradication of eastern monarchies". The examples given are all for Greek dynasties/ families (note Glaukos is a common individual, so that needs to be acknowledged opposite the preference for using about tyrants/monarchs) and for across the eastern and mainland Greek world.

l. 766: my original query remains-- the mysteries teach how to access the special part of the underworld for initiates, as is indicated in l. 745 of the orphic tablet

p. 17 end of n. 24: citation to Edmunds needed; note also in the bibliography the inconsistency in how Edmunds is referenced.

l. 856: typo with Triptolemus

paragraph beginning l. 946: citations needed throughout to the Platonic text

Finally, an apology for not having filled in the recommendations to authors bubbles in the original; I somehow missed them.  

Round 3

Reviewer 1 Report

Comments and Suggestions for Authors

Overall the mss. is improved. The argument and purpose of the sections are now better presented so that the mss. holds together better as a whole.

There are still some issues re: bibliography and chronology that need to be resolved before publication. The chronology issue re: Solon and Anaximander has not been sufficiently addressed; if the author is going to follow Noussia's and Jaeger's chronology in Anaximander's theory being referred to by Solon in fr. 12 and 13, then that needs to be stated. Or the author needs to acknowledge that Solon and Anaximander are working with a shared Ionian philosophical tradition that becomes best known through the work of Anaximander.

Attached is a PDF with comments; please accept this in lieu of typed line by line comments and engage with accordingly.

Comments on the Quality of English Language

Overall fine, some typos. 
